# Scaling Unlocks Broader Generation and Deeper Functional Understanding of Proteins

**Aadyot Bhatnagar**[*,†]    **Sarthak Jain**[*]    **Joel Beazer**[*]    **Samuel C. Curran**[*]

**Alexander M. Hoffnagle**[*]    **Kyle S. Ching**[*]    **Michael Martyn**[*]    **Stephen Nayfach**[*]

**Jeffrey A. Ruffolo**[*,†]                    **Ali Madani**[*,†]

## Abstract

Generative protein language models (PLMs) are powerful tools for designing proteins purpose-built to solve problems in medicine, agriculture, and industrial processes. Recent work has trained ever larger language models, but there has been little systematic study of the optimal training distributions and the influence of model scale on the sequences generated by PLMs. We introduce the ProGen3 family of sparse generative PLMs, and we develop compute-optimal scaling laws to scale up to a 46B-parameter model pre-trained on 1.5T amino acid tokens. ProGen3's pre-training data is sampled from an optimized data distribution over the Profluent Protein Atlas v1, a carefully curated dataset of 3.4B full-length proteins. We evaluate for the first time in the wet lab the influence of model scale on the sequences generated by PLMs, and we find that larger models generate viable proteins for a much wider diversity of protein families. Finally, we find both computationally and experimentally that larger models are more responsive to alignment with laboratory data, resulting in improved protein fitness prediction and sequence generation capabilities. These results indicate that larger PLMs like ProGen3-46B trained on larger, well-curated datasets are powerful foundation models that push the frontier of protein design.[1]

## 1 Introduction

Proteins are ubiquitous molecules that play a central role in most biological processes. They catalyze reactions, contribute to immune function, regulate cellular pathways, and facilitate the transport of other molecules. They enable solutions for multiple industries including therapeutics, diagnostics, agriculture, energy, and manufacturing. Until recently, we have been limited to finding relevant proteins in nature through serendipitous discovery. Techniques to engineer proteins for our desired purposes largely rely on laborious laboratory techniques that randomly mutate these initial hits in hopes of discovering a variant with enhanced function [3, 34]. Relying on these inherently random processes presents a major challenge to the bespoke design of proteins for specific use cases.

The dramatic reduction of DNA sequencing costs has enabled exponential growth in our ability to sample naturally occurring protein sequences that are the product of selective pressures over billions of years. Generative protein language models (PLMs) have emerged as powerful tools to learn the complex distribution of proteins found in nature, and their ability to design highly functional novel proteins has been experimentally verified for a wide range of applications [40, 45, 63, 68, 79, 80, 96].

---

[*]Profluent Bio, Inc.

[†]To whom correspondence should be addressed: {abhatnagar,jeff,ali}@profluent.bio

[1]Code and model weights are available at https://github.com/Profluent-AI/progen3

The trajectory of these models mirrors those found in natural language processing (NLP) research. Earlier works in NLP demonstrated that larger models learn representations that can be fine-tuned to perform downstream tasks more effectively [30, 75, 100, 101], and elucidated the compute-optimal ways to scale up the size of both models and datasets to reap these benefits [43, 51]. Similarly, it has been shown that PLM embeddings implicitly capture notions of protein fitness that can be elicited in both supervised and unsupervised settings [15, 65, 69, 70, 90, 109], and recent work has derived compute-optimal scaling laws for PLMs [19].

However, a number of important questions have remained relatively under-explored in the PLM literature. Despite the rapidly accelerating growth of protein sequence datasets, little work has been done to determine the optimal data distributions for training ever larger PLMs [33]. While evaluations in NLP have shifted towards analyzing the sequences generated by LMs rather than properties of their embeddings [20, 23, 41], it is unclear how scale influences the proteins generated by PLMs, both computationally and experimentally. Finally, post-training strategies to align models with user-defined preferences have gained traction in both domains [74, 88, 106, 110], but the influence of an aligned PLM's scale on its performance has received only limited attention [40].

In this work, we introduce ProGen3, a family of autoregressive generative PLMs that leverages a sparse mixture of experts architecture [32, 56, 85] to improve efficiency while maintaining performance. To train ProGen3, we assemble the Profluent Protein Atlas v1 (PPA-1), a highly curated dataset of 3.4B full-length proteins and 1.1T amino acid tokens, and we optimize the data distribution for pre-training. We determine compute-optimal scaling laws for sparse PLMs and use them to scale ProGen3 up to a 46B parameter model trained on 1.5T tokens from PPA-1. Next, we evaluate for the first time in the wet lab the influence of model scale on the sequences generated by PLMs. We find that larger models generate viable proteins for much wider swaths of sequence space. Finally, we demonstrate both computationally and experimentally that models at all scales can be aligned with laboratory data for improved protein fitness prediction and sequence generation capabilities, and that the larger models receive the greatest lift from alignment. Taken together, these results present a clear case that larger protein language models trained on larger, well-curated datasets are more useful tools for a broad range of protein design challenges.

## 1.1 Related Work

**Mixture of Experts** Scaling up model size is one of the most important ways to improve the performance of deep learning models [43, 51]. However, larger models are also more compute-intensive. Mixture of Experts (MoE) layers leverage sparsity to increase model efficiency [85]. These layers consist of multiple expert sub-networks, and they route each individual token to a different subset of experts. Transformer models trained in the NLP [29, 32, 49, 56], computer vision [77], and protein [90] domains have replaced the feedforward network with a MoE to unlock significant speedups while maintaining performance comparable to a dense model.

**Protein Language Models** Protein language models (PLMs) encompass a wide range of approaches including causal decoders that generate novel proteins from scratch [63, 68, 80] or infill spans in the middle of a protein [15, 86], bidirectional encoders useful for protein understanding [40, 59, 78, 90], and inverse folding models that generate proteins conditioned on a user-specified structure [27, 45, 79], among others. These models are typically pre-trained on large databases of naturally occurring proteins. Compute-optimal scaling laws have been derived for dense PLMs [19], but it is unclear whether they also apply to sparse PLMs.

Limited work has been done to determine the best data distributions for training PLMs, and prior works have largely relied on a range of ad-hoc distributions. Lin et al. [59] and Hayes et al. [40] de-replicated their data at 90% ID and respectively sampled each 50% and 70% ID cluster with equal probability. Nijkamp et al. [68] and Cheng et al. [19] de-replicated their genomic data at 90% ID and their metagenomic data at 30% ID. Chen et al. [15] and Sun et al. [90] de-replicated their genomic data at 90% ID and included at most 10 sequences per 30% ID cluster for their metagenomic data. Meanwhile, Fournier et al. [33] found the best results without any resampling.

**Model Alignment** As language models grow, they acquire increasingly useful emergent properties over the course of unsupervised training [8, 21, 49, 59, 68, 95]. However, out of the box, they are often ill-suited to tasks that their users care about, from generating performant code to design-

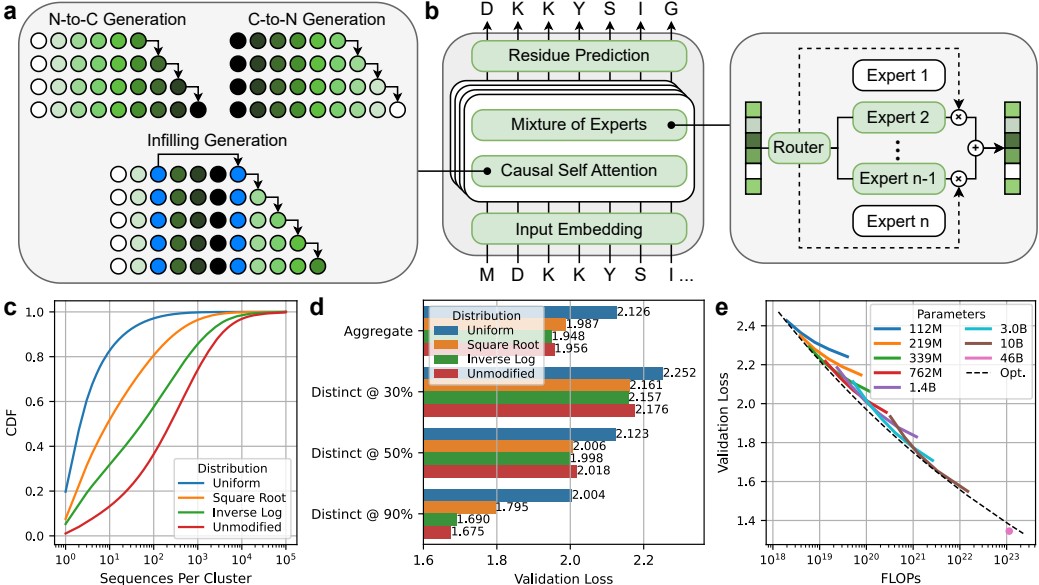

Figure 1: Determining the optimal data distribution and scaling laws to train ProGen3, a sparse generative PLM. (a) ProGen3 can generate proteins from the N- to C-terminal or from the C- to N-terminal. It can also generate spans in the middle of a protein. (b) ProGen3 is an autoregressive transformer with a sparse mixture of experts architecture. (c) Diversity of PPA-1 data distributions as measured by the CDF of 50% ID cluster sizes. (d) Validation losses of 1.4B parameter models trained on 80B tokens from different data distributions. (e) Validation losses of models with 112M to 46B parameters trained on 10B to 1.5T tokens from the Inverse Log distribution. "Opt." indicates the predicted losses for the compute-optimal configurations given by Equation 1.

ing more stable proteins. Alignment algorithms leverage supervised data to direct language model generations to match a user's preferences. These include fine-tuning on a curated set of positive examples [68, 82, 102], reinforcement learning to maximize the user-specified rewards achieved by a language model's conditional generations [22, 110], and direct alignment algorithms that reformulate the reinforcement learning task as an easier supervised learning task [71, 74]. For PLMs, direct alignment algorithms have improved models for inverse folding [40, 106], fitness prediction [109], function-guided generation [40, 88], and binder design [48, 67].

## 2 ProGen3

### 2.1 Architecture

ProGen3 is a family of autoregressive (AR) transformer based PLMs with a sparse mixture of experts (MoE) architecture that only activates 27% of its parameters per forward pass [32, 56, 85, 99] (Figure 1b). Models span 8 sizes ranging from 112M to 46B parameters, and all models have a context length of 8192 tokens. See Appendix B.1 for more details. Like prior work in NLP [29, 49], we find that for a fixed compute budget, sparse models meaningfully outperform dense ones (Appendix B.2).

A standard AR PLM enables *causal language modeling* (CLM), where users can generate proteins one amino acid at a time either from the N- to C-terminal, or from the C- to N-terminal [63, 68]. In addition to CLM, ProGen3 also performs *generalized language modeling* (GLM) to infill spans in the middle of a protein [6, 15, 35, 75, 83, 92]. It does so by replacing target spans with special sentinel tokens and placing those spans (preceded by their corresponding sentinel tokens) at the end of the sequence (Figure 1a). This capability allows users to redesign domains in the middle of a protein while attending to all surrounding residues. See Appendix B.3 for more details.

## 2.2 Training Data

To train ProGen3, we curate the Profluent Protein Atlas v1 (PPA-1), a dataset of 3.4B proteins and 1.1T amino acid tokens. PPA-1 draws from a wide range of genomic and metagenomic sources, and we apply multiple layers of quality filters to ensure the dataset is appropriate for pre-training a generative PLM. In particular, we exclude all protein fragments so we can train only on full protein sequences. PPA-1 is of similar scale and diversity to the OMG dataset [24]. It has a similar number of sequences as ESM3's dataset, but ESM3's dataset does include protein fragments [40]. It is also considerably larger than UniRef [91] and BFD [50, 87]. See Appendix A for more details.

While protein datasets are rapidly expanding, they have biases that influence the quality of PLMs trained on them [4, 31, 103]. Therefore, we aim to more rigorously study the impact of the training distribution, an aspect which has been relatively underexplored in prior PLM literature (Section 1.1). We consider 4 different schemes of balancing data diversity over 50% ID clusters, which we refer to (in descending order of diversity) as Uniform, Square Root, Inverse Log, and Unmodified (Figure 1c). In each of these distributions, we respectively sample a 50% ID cluster of size $n$ with probability proportional to $1$ (equal probability), $\sqrt{n}$, $n/(1+\log n)$, and $n$ (the natural sequence distribution). These distributions are listed in descending order of diversity: 80% of sequences sampled from these distributions reside in clusters of size at most 8, 94, 591, and 1471, respectively. After sampling a 50% ID cluster, we sample a 90% ID sub-cluster from the same distribution, before sampling a sequence from that 90% ID sub-cluster uniformly at random.

To measure model generalization, we construct validation sets distinct from our training data at 30%, 50%, and 90% ID. Each x% ID validation set consists of 2.5M sequences distributed uniformly between X% ID clusters of sizes 1-10, 11-100, and 101-1000. The average loss thus avoids over-weighting highly represented parts of protein space and more accurately measures out-of-distribution generalization. We also average the losses on these sets to compute an aggregate validation loss.

Figure 1d shows the validation losses achieved by 1.4B parameter models trained for 80B tokens sampled from each distribution. For proteins close to our training data – distinct at 90% ID – training on the Unmodified distribution yields the best performance. The Inverse Log distribution is a close second, while the other two are considerably worse. However, training on the Inverse Log distribution yields the lowest loss for out-of-distribution sequences that are distinct from the training data at 50% and even 30% ID. We therefore train all subsequent models on the Inverse Log distribution.

Interestingly, training on the Uniform distribution, which is similar in spirit to de-replicating the training data at 50% ID, consistently obtains the worst performance on all validation sets. This suggests that PLMs can learn important signals from the frequencies with which related proteins occur in the data. However, some rebalancing can improve out-of-distribution generalization.

## 2.3 Scaling Up to a 46B Parameter Model

Given a fixed computational budget, scaling laws describe the optimal way to allocate pre-training resources between model size and dataset size [19, 43, 51]. They do so by predicting a model's loss $L$ as a function of the number of parameters $N$ and the number of tokens in the training dataset $D$. We fit a variant of the equation proposed by Kaplan et al. [51], $L(N,D) = \left(AN^{-\alpha/\beta} + BD^{-1}\right)^{\beta} + C$, to the validation losses achieved by models spanning a wide range of parameter counts and dataset sizes. This equation can then be manipulated to yield a power law $N_{\text{opt}}(D) \propto D^{\beta/\alpha}$ which tells us the compute-optimal model size to pre-train on a dataset with $D$ tokens.

We train models with 112M to 10B parameters for 10k to 500k steps (10B to 1T tokens, depending on batch size; see Appendix B.1). Because we use a cosine learning rate schedule, we train a separate model for each dataset size [43]. We plot their validation losses in Figure 1e and find

$$N_{\text{opt}}(D) = (2.462 \times 10^{-7})D^{1.479}. \tag{1}$$

This result for sparse AR PLMs agrees closely with Cheng et al. [19], who report that $N_{\text{opt}}(D) \propto D^{1.370}$ for dense AR PLMs. Equation 1 predicts that the optimal allocation for $1.1 \times 10^{23}$ FLOPs is to train a 90B parameter model for 753B tokens. This model is predicted to achieve a validation loss of 1.376. We make the practical decision to instead train a 46B parameter model for 1.5T tokens because it is predicted to achieve a slightly higher validation loss of 1.397 but can be used for inference on 4xA100_40GB GPUs.

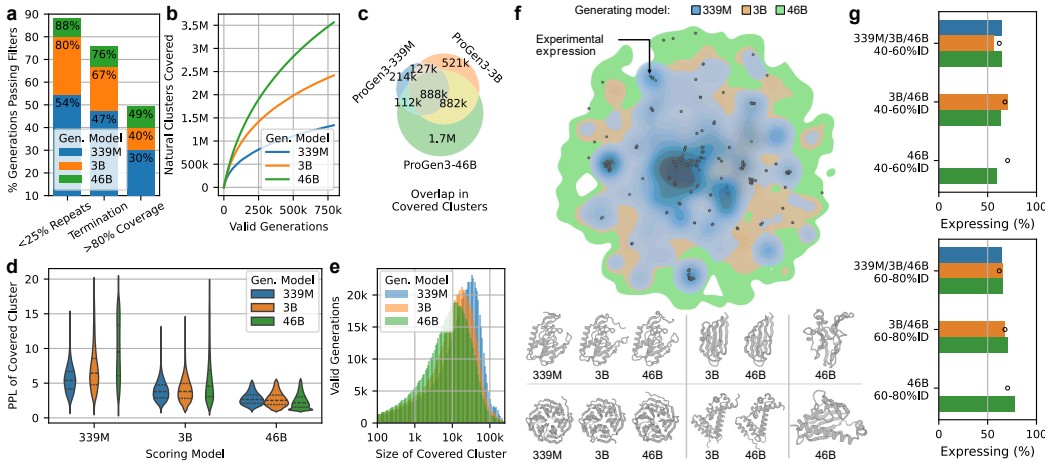

Figure 2: Larger models generate viable proteins for a much more diverse set of families than smaller models. (a) Larger models generate more *valid* sequences that pass a series of quality filters. (b) Valid generations from larger models cover a wider array of naturally occurring 30% ID clusters. (c) The clusters covered by larger models are largely a superset of those covered by smaller models. (d) Larger models generate sequences from clusters whose natural members are assigned high perplexity by smaller models, but not vice versa. (e) Larger models are better able to generate sequences from smaller clusters. (f) Visualization of the clusters selected for experimental characterization, accompanied by ESMFold predicted structures for selected generations. (g) *In vitro* expression rates of model generations for different groups of clusters. Black circles indicate natural protein expression rates for the same clusters.

Interestingly, this 46B parameter model achieves a validation loss of 1.345, which lies below the predicted compute-optimal frontier (Figure 1e, bottom right). We had to increase the warmup period to stabilize the training for this model, which could explain the discrepancy, since a longer warmup improved the final training and validation losses of a 1.4B parameter model trained for 80B tokens (Appendix B.4).

## 3 Computational and Experimental Analysis of Model Generations

We have shown that scaling up the computational resources to pre-train PLMs predictably improves their validation losses for proteins both near and distant to their training data. In other words, larger models better represent the naturally occurring distribution of proteins. However, the implications of scaling on the generative capabilities of PLMs have largely gone unexplored by prior works. We focus on three models separated by three orders of magnitude in pre-training compute: ProGen3-339M (trained on 200B tokens with $1.2 \times 10^{20}$ FLOPs), ProGen3-3B (trained on 500B tokens with $2.6 \times 10^{21}$ FLOPs), and ProGen3-46B (trained on 1.5T tokens with $1.1 \times 10^{23}$ FLOPs). We also evaluate ProGen2-XL [68], a previous-generation dense autoregressive model with 6.4B parameters, and ProGen3-3B-200B, a variant of ProGen3-3B trained on only 200B tokens.

### 3.1 Larger PLMs generate more diverse proteins that express *in vitro*

From each ProGen3 model, we respectively generate 4M, 3M, and 2M sequences unconditionally using top-$p$ sampling [44] with $p = 0.95$ and temperatures $T \in [0.5, 1]$. We also sample 4M generations from ProGen2-XL using the same hyperparameters.

We then apply a series of quality filters to extract "protein-like" sequences. Because language models often generate repetitive sequences [44, 105], we first remove all sequences consisting of >25% low-complexity regions [36]; 95.5% of PPA-1 also passes this filter. We also require that generations include the appropriate termination token, and that they have >80% alignment coverage to any natural sequence in PPA-1 [2, 80]. Larger ProGen3 models generate more *valid* sequences that pass each of these quality filters, and scaling up from 339M to 3B especially reduces the number of repet-

itive generations (Figure 2a). Interestingly, only 4% of the sequences generated by ProGen2-XL are valid, compared with 30%, 40%, and 49% from ProGen3-339M, 3B, and 46B, respectively.

Next, we align each valid generation to the 30% ID cluster representatives in PPA-1 and say that it *covers* a cluster if it is >30% ID to the cluster representative with >90% alignment coverage. Note that a single generation can cover multiple clusters because a sequence can be >30% ID to multiple cluster representatives that are <30% ID to each other. Larger models generate sequences from a more diverse set of clusters than smaller models (Figure 2b). Moreover, the clusters covered by smaller models are almost entirely covered by larger models, while larger models generate sequences from many clusters that are not covered by smaller models (Figure 2c). Amongst its valid generations, ProGen2-XL covers 18% fewer clusters than ProGen3-3B despite being a larger model.

We also find that for ProGen3, model size has more influence than dataset size, further corroborating our scaling laws (Equation 1). ProGen3-3B-200B generates 38% valid sequences, compared to 30% for ProGen3-339M (200B tokens) and 40% for ProGen3-3B (500B tokens). Meanwhile, ProGen3-3B covers 81% more clusters than ProGen3-339M, but only 14% more than ProGen3-3B-200B.

To determine whether sampling more sequences would allow smaller ProGen3 models to generate from the clusters covered by larger models, we compute the average perplexities of the naturally occurring proteins in those clusters. If a given model assigns high perplexity to a cluster's naturals, it is unlikely to generate a sequence from that cluster. Figure 2d shows that larger models routinely generate from clusters that smaller models assign high perplexity, but not vice versa. This implies that larger models can cover all the clusters that smaller models generate from, but that some clusters covered by larger models are out-of-distribution for smaller models. This trend is partially explained by the fact that as models scale up, they become better at covering smaller clusters (Figure 2e).

We now seek to characterize the viability of these generated proteins in the lab. It is infeasible to directly measure protein function for a wide range of families with potentially unknown functions. Therefore, we instead perform the split-GFP *E. coli* protein expression assay [11, 12] (Appendix D) to measure soluble protein abundance, which depends on factors including mRNA abundance and stability, translational yield, thermodynamic folding equilibrium, kinetic stability, susceptibility to proteases, aggregation propensity, and toxicity [5, 60]. This assay correlates well with protein expression assayed by direct methods such as SDS-PAGE and Western blotting, is amenable to high throughput assay [61], and has been used to benchmark generative protein models [47]. All of these qualities are important components of a protein's fitness.

Given that increased diversity is one of the clearest impacts of scaling up PLMs, we select for experimental characterization 42 clusters that all 3 models generated from, 62 clusters that ProGen3-3B and ProGen3-46B generated from, and 45 clusters that only ProGen3-46B generated from (Figure 2f). From each cluster, we select one sequence per model that is 40-60% ID to a natural sequence in PPA-1, and one sequence that is 60-80% ID. For each (cluster, %ID bucket) pair, the selected sequences have the lowest perplexity according to the model that generated them. We also include as controls two random natural proteins for each cluster, and we ensure that the sequences in the selected clusters span a wide range of lengths (75-300aa) and structural diversity.

Figure 2g reports the cluster-averaged percentage of sequences that showed soluble expression *in vitro*. All models attain similar rates of expression, but larger models do so for a wider range of clusters. As one might expect, generated proteins that are 40-60% ID to a natural tend to have lower rates of expression than those that are 60-80% ID. Interestingly, for all models, the generated proteins expressed at rates similar to random naturals from the clusters they reside in.

## 3.2 PLMs generate viable proteins outside natural sequence space

Having demonstrated that proteins generated by PLMs residing in sequence clusters alongside naturals are broadly viable, we next turn to sequences without a natural reference point, i.e. they are <30% identity to any protein in PPA-1 (or could not be aligned). From the same set of unconditional generations, we select 10 each that have mostly-alpha, mostly-beta, and mixed alpha/beta secondary structure compositions (as predicted by ESMFold [59]) from ProGen3-339M, ProGen3-3B, and ProGen3-46B, for a total of 90 sequences. We consider the rate of expression using the split-GFP assay (Figure 3a), as well as the expression levels relative to our split-GFP positive control (Figure 3b). We find that all models are comparable in their ability to generate sequences that express *in vitro*, but generations from larger models generally have higher expression.

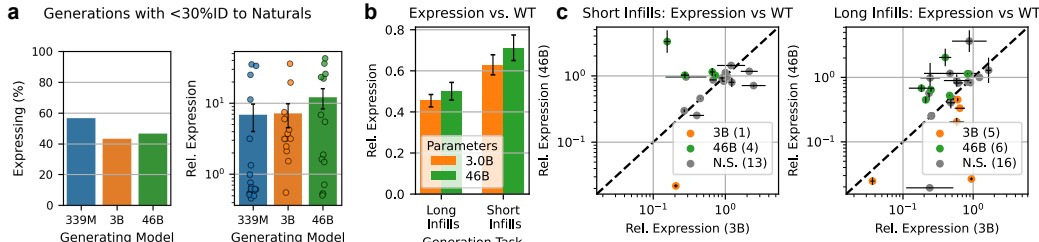

Figure 3: Protein language model generate novel proteins and infill effectively. (a) Expression rates (left) and levels relative to positive control (right) for generated proteins with less than 30% identity to any natural sequence. (b) Expression levels of infilled sequences relative to respective scaffold proteins for long- and short-span infilling tasks. (c) Comparison of relative expression levels for infills generated for the same spans by models with 3B and 46B parameters.

### 3.3 Larger PLMs are better infillers

ProGen3's infilling capabilities enable redesigning key segments in the middle of a proteins sequence. To test these capabilities, we select nine therapeutically or industrially relevant proteins 100-400aa in length and with some precedent for compatibility with recombinant *E. coli* expression. For each protein, we use ProGen3-3B and ProGen3-46B to infill sequences for two randomly selected short spans (20% of the protein's length) and two randomly selected long spans (50% of the protein's length). We also randomly select nine additional cytosolic proteins between 100aa and 300aa from the *E. coli* proteome and use the two models to infill sequences for one short span and one long span.

For each span, we select from each model the two infills with the lowest perplexity according to the model that generated them. We experimentally determine the expression levels of two selected sequences from each model, as well as the expression of the corresponding natural protein, using our split-GFP assay.

11 of the 18 natural proteins come from the *E. coli* proteome and may therefore be expected to express well in our split-GFP assay. While infilled sequences have reduced expression on average compared to their corresponding natural protein (Figure 3b), there are nonetheless examples of infilled sequences with improved expression over their natural counterpart. Overall, ProGen3-46B slightly outperforms ProGen3-3B, both by the average expression across all infills (Figure 3b) and by the relative expression for a given infilled span (Figure 3c). This suggests that larger models may be more adept at understanding the context and constraints present in a given sequence for infilling.

## 4 Aligning Protein Language Models to Laboratory Data

Zero-shot fitness prediction is often used to evaluate PLMs pre-trained on evolutionary data. Performance is typically measured by the Spearman correlation $\rho$ between model likelihoods and experimentally determined fitness values for the proteins in a deep mutational scan (DMS) dataset [42, 65, 69]. However, due to correlations present in evolutionary dynamics, a PLM's task of faithfully capturing the natural protein distribution can be at odds with zero-shot fitness prediction [103].

Indeed, despite the fact that scaling up model size decreases the validation loss of proteins highly distant from their training data and allows them to generate a much greater diversity of expressing proteins, we find that models larger than 3B parameters often perform worse on the ProteinGym benchmark [69] (Figure 4a). This finding strengthens with more extensive experiments the hypotheses of Weinstein et al. [103] and Gordon et al. [38], that beyond a certain threshold, better estimators of the natural protein distribution can be worse fitness predictors.

Instead, we find that a key advantage of larger models lies in their latent abilities which can be elicited through supervised training. To this end, we use iterative reasoning preference optimization (IRPO, [71]) to align ProGen3 likelihoods with experimentally measured protein properties including activity, binding, organismal fitness, and stability. See Appendix C.1 for more details.

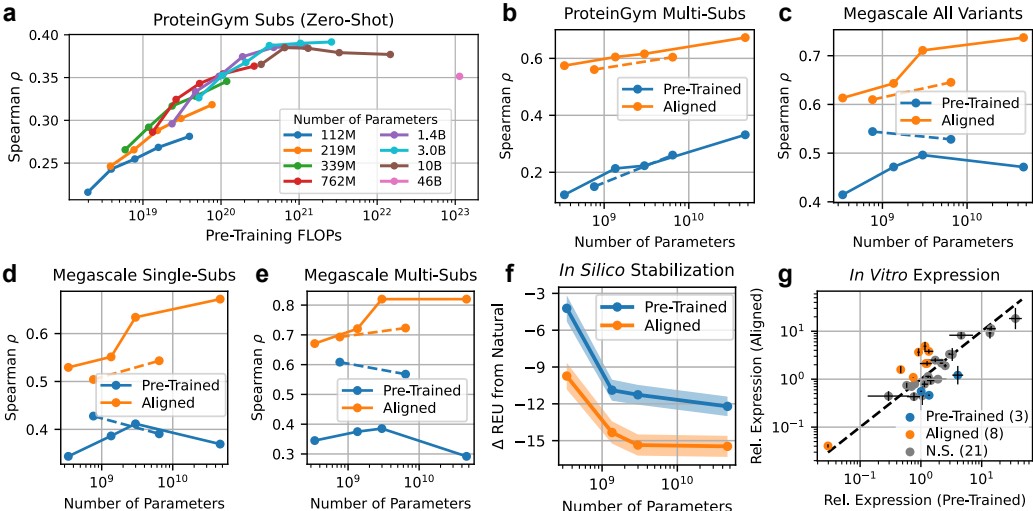

Figure 4: Aligning a model with laboratory data improves its utility for practical design tasks, and larger models are more amenable to alignment. Solid lines indicate ProGen3, while dashed lines indicate ProGen2. (a) Increasing model scale beyond a certain point degrades zero-shot fitness prediction. (b) Aligning models on a local mutational landscape improves their fitness prediction capabilities for much more distant variants. ProGen3 outperforms ProGen2. (c) Aligning models on protein stability datasets allows them to perform out-of-distribution stability prediction. ProGen3 outperforms ProGen2. (d) Stability prediction performance on single-substitution variants. (e) Stability prediction performance on multi-substitution variants. (f) Average change in Rosetta energy (REU) between natural proteins and sequences generated using those proteins as prompts. Alignment and model scale both improve the *in silico* stability of generated sequences. (g) Aligning ProGen3-46B on stability data improves the *in vitro* expression of its generations in many cases.

## 4.1   Supervised Fitness Prediction

First, we demonstrate how alignment can accelerate a directed evolution campaign by training Pro-Gen3 models on a local mutational landscape and performing fitness prediction on more distant variants. We identify all assays in ProteinGym [69] with at least 3 mutations, and we train on all variants at most $k$ mutations from the wild type, where $k$ is the smallest number required for the train split to exceed 500 sequences. To ensure that the train and test splits contain proteins of similar fitness, we require that the total variation distance between the train and test distributions of fitness scores be less than 1. These filters yield 8 assays that measure diverse functional attributes for a wide range of proteins (Supp. Table 8).

With as few as 500 experimental measurements of single-substitution protein variants, aligned models learn generalizable concepts of fitness that allow them to accurately rank proteins that are tens of mutations away from the wild type. The overall fitness prediction performance of aligned models increases with their size (Figure 4b, Supp. Figure 6). An aligned ProGen3-46B ($\rho = 0.673$) outperforms KERMUT [39] ($\rho = 0.628$) and obtains similar performance to ConFit [109] ($\rho = 0.679$). Both baselines are state-of-the-art methods that leverage inductive biases specifically designed to predict the effects of substitutions. In contrast, ProGen3 can also generate novel proteins, and our IRPO formulation can accommodate insertions and deletions as well as substitutions.

Next, we align ProGen3 models with folding free energy measurements ($\Delta G$) from the Megascale protein stability dataset [97], which consists of almost 800,000 single- and double-mutation variants of 479 protein domains. We evaluate how well the models learn a universal concept of stability that generalizes to proteins highly distinct from the training data. Like Widatalla et al. [106], we use FoldSeek `easy-cluster` [98] with 50% alignment coverage to structurally cluster the domains. We hold out 5% of clusters for validation and 5% for testing. Additionally, we only train on single-substitution variants.

As before, increasing model size uniformly improves the stability prediction performance of an aligned model (Figure 4c-e). Using a similar train/val/test split, ProteinDPO [106], a similar method that additionally leverages structural information, obtains comparable overall performance ($\rho = 0.72$) to the aligned ProGen3-46B ($\rho = 0.737$, Figure 4c). However, despite only being trained on single-substitution variants, our sequence-only method is a much better stability predictor for multi-substitution variants, achieving $\rho = 0.820$ compared to ProteinDPO's $\rho = 0.468$ (Figure 4e).

Finally, across all fitness prediction tasks, we find that for a fixed parameter count, applying IRPO to ProGen3 outperforms applying IRPO to ProGen2 (Figure 4b-e), but ProGen3 is much more efficient due to its sparse MoE implementation. This further highlights the importance of the improved architecture and pre-training data distribution that differentiate ProGen3 from ProGen2.

## 4.2 Sequence Generation

Encouraged by these results, we now evaluate the generative capabilities of stability-aligned models. From the set tested in Section 3.1, we select as prompts 32 structurally diverse natural proteins that are 98-282 amino acids long (considerably longer than the domains in the Megascale dataset, which are 40-72 amino acids long). We generate 50,000 sequences per model per prompt subject to the quality filters described in Figure 2a, either the C-terminal 75% conditioned on the N-terminal 25%, or the N-terminal 75% conditioned on the C-terminal 25%. For each (model, prompt) combination, we select the 100 sequences passing these quality filters that have the lowest perplexity according to the model that generated them.

As an initial evaluation, we perform an *in silico* characterization of protein stability. For each batch of generated sequences and the corresponding natural sequence prompt, we predict the structure with ESMFold [59] and perform four cycles of minimization (`MinMover`) and relaxation (`FastRelax`) across five independent trajectories in PyRosetta [14]. To evaluate the effects of model scale and alignment on the stability of generated proteins, we calculate the difference in the minimum Rosetta energy achieved for each sequence relative to the corresponding natural (lower is more stable). We find that larger models generally produce more stable proteins and that alignment further stabilizes proteins across model scales (Figure 4f).

Finally, prior work has found that the split-GFP assay correlates positively with protein stability [5, 60]. To determine whether alignment on protein stability data actually improves the stability of generated proteins, we select for each prompt the two lowest-perplexity sequences from both pre-trained and stability-aligned ProGen3-46B models, and we characterize them with the split-GFP assay. In Figure 4g, we plot for each prompt the average expression of each model's generations, normalized by the expression of the wild type protein used to construct each prompt. Using a paired-sample Welch's $t$-test [104] with $p < 0.01$, we find that alignment improves *in vitro* expression for 8/32 prompts, degrades it for 3/32 prompts, and has no significant effect for 21/32 prompts.

We thus verify both computationally and experimentally that alignment not only improves a ProGen3 model's stability prediction capabilities, but also improves the stability of the sequences it generates. Moreover, alignment imbues these models with a concept of protein stability that generalizes to proteins that are structurally dissimilar to and considerably longer than any of the sequences found in the alignment dataset. Since alignment also improves out-of-distribution fitness prediction more broadly, we expect that it can be applied to improve models' abilities to generate sequences that optimize a diverse array of functional attributes, and that larger models will see greater gains.

## 5 Discussion

In this work, we introduce ProGen3, a family of sparse, optimally-scaled generative protein language models (PLMs) trained on one of the largest high-quality protein datasets constructed to date. We systematically demonstrate the importance of curating appropriate data distributions for pre-training PLMs, and we use our findings to train a 46B parameter instantiation of ProGen3 on 1.5T tokens. To our knowledge, ProGen3-46B is the largest sparse PLM yet, and it required 5-10x less compute to pre-train than dense frontier models of a comparable scale [15, 40].

We perform wet lab studies on the relationship between model scale and the fitness of proteins generated by PLMs, and we find that larger models generate viable proteins for a much wider diversity of protein families. Thus, an immediate consequence of model scaling is the ability to better rep-

resent rare protein families. Larger models' more general representations also improve their ability to redesign a protein of interest for improved expression, even if that protein is well-represented in the training data. Finally, we show both computationally and experimentally that larger models reap the greatest benefits to their protein fitness prediction and sequence generation capabilities when we align them with laboratory data. In particular, an aligned ProGen3-46B consistently matches or beats state-of-the-art supervised fitness prediction methods while being a far more flexible model.

We have thus demonstrated that larger generative PLMs are more useful tools for a wide range of real-world protein design tasks. Continued model scaling can leverage the exponentially growing amounts of protein sequence data while employing more sophisticated implementations of sparsity to remain highly efficient [29]. However, our results show that ProGen3-46B is already well poised to advance the vision of designing bespoke proteins for use cases that include drug discovery, enzyme engineering, and industrial processes.

## Safety and Ethics

Computational protein design carries the dual potential to accelerate the development of novel therapeutics and other society-improving molecules, while providing parallel capabilities for nefarious uses, such as engineering of bioweapons. When bolstered by current and future iterations of generative AI, these capabilities are heightened and expected to grow further. The global protein design community has begun to establish appropriate regulations and guidelines towards the continued beneficial development and application of these technologies. We support having a set of community values, guiding principles, and commitments for the responsible development of AI for protein design (https://responsiblebiodesign.ai). Gene synthesis represents a critical step in the actualization of designed protein sequences. The International Gene Synthesis Consortium (IGSC) unites major gene synthesis providers under a commitment to screen all incoming orders against known pathogens and potentially dangerous sequences. As a concrete step towards safe application of protein design technology, all gene synthesis work in support of the present study was performed with IGSC members. For all protein design projects, we urge researchers to maintain ethical oversight throughout project initiation, experimental characterization, and subsequent deployment phases to ensure safety and avoid unintended harmful outcomes. For the current models described in this paper for release, we find the benefit of model accessibility to greatly outweigh any theoretical risks.

## Competing Interests

The authors are current or former employees, contractors, or executives of Profluent Bio, Inc. and may hold shares in Profluent Bio, Inc. Funding for this work was provided by Profluent Bio, Inc.

## Acknowledgments

We would like to thank Adeline Chen and Rachel Lee for developing our high-throughput lab infrastructure, Eleanor Dunietz for developing our lab informatics systems, and Joseph Gallagher and Peter Cameron for overarching support.

## Author Contributions

**Data:** J.B., S.N.

**Pre-Training:** A.B., S.J.

**Alignment:** A.B.

**Analysis and Experimental Design:** A.B., J.A.R., S.C.C., A.M.H., S.J., A.M.

**Split-GFP Experiment:** S.C.C., K.C., M.M.

**Open Source Development**: S.J.

**Overall Scientific Direction**: A.M., A.B., J.A.R.

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

# A   Profluent Protein Atlas

To train ProGen3, we curated the Profluent Protein Atlas v1 (PPA-1). Table 1 summarizes the composition of the dataset. We collected genomic and metagenomic datasets from IMG/M [16], ENA [55], and NCBI GenBank [81], and we performed protein coding prediction using `prodigal-gv` [13, 46]. We discarded contigs shorter than 1,000 bp as well as contigs with coding density <65% to remove mispredicted proteins from eukaryotic sequences [73]. These were combined with proteins from UniRef [91], NCBI NR [81], and five additional metagenome and eukaryotic-focused databases, resulting in a total of 12.2B proteins. All proteins were subject to additional filters to remove proteins with >8000 amino acids, protein fragments, and proteins with invalid amino acids, k-mer repeats [80], or >50% low complexity regions [36]. This resulted in a final filtered set of 3.4B proteins.

On this filtered set, we performed hierarchical clustering at 90%, 50%, and 30% amino acid identity using DIAMOND [9]. At each clustering step, we used the representative sequences from the previous round (e.g., 50% identity clustering was performed on the 90% identity cluster representatives). Clustering at each level was conducted iteratively across four sensitivity settings, where the representatives from one setting were clustered further at the next. The number of sensitivity levels varied by identity threshold: at 90% identity, we applied only `--linclust`, while at 50% and 30% identity, we used `--linclust`, `--fast`, `--default`, and `--more-sensitive`. This pipeline yielded 1.2B 90% ID clusters, 183M 50% ID clusters, and 113M 30% ID clusters.

We compare PPA-1 to other protein datasets in Table 2. After filtering and clustering, PPA-1 is of similar scale and diversity to the OMG dataset [24]. It is of similar scale to the dataset used to train ESM3 [40], but it does not include any protein fragments while ESM3's dataset does. It is also considerably larger than resources like UniRef [91] and BFD [50, 87].

| Data Source | Type | Proteins (Total) | Proteins (Filtered) | Percentage |
|---|---|---|---|---|
| IMG/M [16] | Metagenomic | 5,313,407,975 | 1,402,061,253 | 41.28% |
| ENA [55] | Metagenomic | 3,324,676,018 | 946,087,610 | 27.85% |
| NCBI-nr [81] | Genomic | 542,683,057 | 471,482,093 | 13.88% |
| GenBank [81] | Genomic | 455,377,658 | 362,034,076 | 10.66% |
| UniRef90 [91] | Genomic | 170,669,877 | 144,914,015 | 4.27% |
| SRC [87] | Metagenomic | 2,020,821,851 | 60,061,204 | 1.77% |
| MMETSP [52] | Metagenomic | 28,029,947 | 3,837,925 | 0.11% |
| SoilEuk [7] | Metagenomic | 5,366,252 | 3,450,399 | 0.10% |
| MERC [87] | Metagenomic | 291,261,719 | 1,994,696 | 0.06% |
| MetaEuk [57] | Metagenomic | 6,310,729 | 258,456 | 0.01% |
| Total | — | 12,158,605,083 | 3,396,181,727 | 100% |

Table 1: Composition of PPA-1.

| Dataset | Fragments Removed | 30% ID clusters | 50% ID Clusters | Proteins |
|---|---|---|---|---|
| UniRef [91] | ✗ | — | 69M | 454M |
| BFD [50, 87] | ✗ | 66M | — | 2204M |
| ESM3 [40] | ✗ | — | — | 3143M |
| OMG [24] | ✓ | — | 207M | 3300M |
| PPA-1 (ours) | ✓ | 113M | 183M | 3396M |

Table 2: Comparison of PPA-1 to other large protein datasets.

# B   Pre-Training

## B.1   Implementation and Hyperparameters

All ProGen3 models are transformer based decoder-only models[99] with a context length of 8192 tokens and Transformer++ optimizations [95]: feedforward networks with SwiGLU activations

[28, 84], rotary positional embeddings (RoPE [89]), grouped query attention [1], and pre-layer normalization [107] using RMSNorm [108]. For RoPE, we use $\theta = 10^5$. Unless otherwise specified, we replace all feed-forward layers with a Mixture of Experts (MoE) that activates 2 experts (out of 8 total) for each token [32, 85]:

$$\text{MoE}(x) = \sum_{i=1}^{8} \text{Softmax}(\text{Top2}(W_g x))_i \cdot \text{FFN}_i(x), \tag{2}$$

where $x \in \mathbb{R}^d$ is a single token embedding, $W_g \in \mathbb{R}^{8 \times d}$ is a linear routing layer, and $\text{FFN}_i : \mathbb{R}^d \to \mathbb{R}^d$ is an expert sub-network. We regularize our language modeling loss with the load balancing loss used by SwitchTransformer [32] multiplied by a weight $\lambda = 0.05$.

All models are trained using the AdamW optimizer [53, 62] with $\beta_1 = 0.9$, $\beta_2 = 0.95$, and BF16 mixed precision [66]. After an initial warmup period, we decay the learning rate to 10% of its peak value following a cosine schedule. We leverage fully sharded data parallel training [76] and gradient checkpointing [18] for memory-efficient distributed training.

We implement our models using PyTorch [72]. To improve efficiency, we use `FlashAttention2` [26] and `Megablocks` [37] for our attention and MoE layers, respectively. Finally, we orchestrate training and data loading with MosaicML's `composer` [93] and `streaming` [94] libraries, respectively. We trained all models on H100s hosted by MosaicML/Databricks. Pre-training ProGen3-46B took approximately 17 days on a cluster of 256xH100.

Table 3 describes our model configurations and pre-training hyperparameters in more detail.

| Params (Active) | Layers | $d_{\text{model}}$ | Attn Heads | $d_{\text{FFN}}$ | LR | WD | BSZ | WU |
|---|---|---|---|---|---|---|---|---|
| 112M  (32M) | 10 | 384 | 6 | 1152 | 5e-4 | 5e-6 | 1M | 2000 |
| 219M  (63M) | 11 | 512 | 8 | 1536 | 5e-4 | 5e-6 | 1M | 2000 |
| 339M  (98M) | 17 | 512 | 8 | 1536 | 5e-4 | 5e-6 | 1M | 2000 |
| 762M (221M) | 17 | 768 | 12 | 2304 | 5e-4 | 5e-6 | 1M | 2000 |
| 1.4B (393M) | 17 | 1024 | 16 | 3072 | 5e-4 | 5e-6 | 1M | 2000 |
| 3.0B (866M) | 24 | 1280 | 16 | 3840 | 5e-4 | 5e-6 | 1M | 2000 |
| 10B   (2.7B) | 24 | 2048 | 16 (8 KV) | 8192 | 5e-4 | 5e-6 | 2M | 2000 |
| 46B (12.6B) | 32 | 4096 | 32 (8 KV) | 14336 | 3.2e-4 | 4e-6 | 3M | 10000 |

Table 3: Model configurations and pre-training hyperparameters. BSZ is the number of tokens per batch. WU is the number of warmup steps and is kept constant regardless of total training duration.

## B.2    Mixture of Experts Performance

To validate the performance of sparse architectures, we train three models for 80B tokens: a dense 1.4B parameter model, a sparse 1.4B parameter model, and a sparse 3B parameter model. In Table 4, we find that the dense 1.4B parameter model slightly outperforms the sparse 1.4B parameter model. However, the sparse 3B parameter model significantly outperforms both while requiring about half the FLOPs of the smaller dense model.

| Architecture | Parameters | Active Parameters | FLOPs | Validation Loss |
|---|---|---|---|---|
| Dense | 1,386,084,352 | 1,386,084,352 | $1.10 \times 10^{20}$ | 1.920 |
| Sparse | 1,355,871,232 | 393,174,016 | $3.14 \times 10^{19}$ | 1.948 |
| Sparse | 2,989,920,000 | 866,369,280 | $6.93 \times 10^{19}$ | 1.849 |

Table 4: Validation loss of dense and sparse architectures trained for 80B tokens. For a fixed number of parameters, dense models slightly outperform sparse ones. For a fixed FLOPs budget, sparse models outperform dense ones.

## B.3    Training Tasks

Training Protein Language Models with a standard autoregressive modeling objective supports generating protein sequences from the N-to-C terminal or C-to-N terminal, conditioned on the previous

| | Validation PPL (Distinct @50%) | | | |
|---|---|---|---|---|
| Hyperparameter | CLM | GLM ($L = 10$) | GLM ($L = 50$) | GLM ($L = 200$) |
| **Infilling to CLM Ratio** | | | | |
| 0 | 8.40 | – | – | – |
| 1:10 | 8.82 | 20.86 | >100 | >100 |
| 1:4 | 8.98 | 14.84 | >100 | >100 |
| 1:2 | 9.13 | **12.05** | >100 | >100 |
| 1:1 | 9.41 | 12.47 | >100 | >100 |
| **Mask Fractions and Span Lengths** | | | | |
| $L \leq 10, f \leq 0.15$ | 9.13 | 12.05 | >100 | >100 |
| $L \leq 30, f \leq 0.25$ | 9.08 | 9.64 | 49.16 | >100 |
| $L \leq 70, f \leq 0.50$ | 9.09 | 9.71 | 9.68 | 9.44 |
| $L \leq 200, f \leq 0.50$ | **9.02** | **9.54** | **9.34** | **8.91** |
| **Positional Encoding Strategy** | | | | |
| Naive | 8.98 | 9.47 | 9.30 | 8.87 |
| Position-Preserving | 8.99 | 9.08 | **8.86** | 8.56 |
| Fuzzy Position-Preserving | **8.90** | **8.30** | 8.87 | **8.56** |

Table 5: Validation Loss on CLM and Single Span Infilling Tasks for various infilling setups.

residues in the sequence [63, 68]. This formulation precludes settings where we only want to generate a segment in the middle of the protein while keeping both prior and subsequent amino acids fixed. To support this, we augment the ProGen3 model training with an infilling or *generalized language modeling (GLM)* learning objective, taking inspiration from multiple prior works in NLP [6, 35, 75, 92] and Protein Language Modeling [15, 83].

**Infilling Training Details** During ProGen3 training, we used the infilling objective for 1/3 of the sequences seen by the model (i.e., the GLM to CLM ratio is 1:2). For each such case, we sample $N$ spans from the sequence, up until a maximum length fraction $f$. The span length of each span is sampled from a truncated mixture of Gaussians, where we assign equal probability to each of $\mathcal{N}(10, 5)$, $\mathcal{N}(30, 10)$, $\mathcal{N}(70, 20)$, $\mathcal{N}(200, 50)$, and $\mathcal{N}(400, 100)$, denoted as $\mathcal{N}(\mu, \sigma)$. The maximum length fraction $f$ is itself sampled from the set $\{0.15, 0.25, 0.5, 0.8\}$ with probabilities $\{0.28, 0.3, 0.28, 0.14\}$ respectively.

For each span, we sample the start position uniformly from the protein sequence and we replace the corresponding span of residues (span start, span start + span length) with a single sentinel token <span_i>. At the same time, the same sentinel token is appended to the end of the sequence, followed by the residues in the span being replaced and finally an end of span token. The constructed sequence is then used to train the model with a causal attention mask and autoregressive modeling loss over the whole sequence. Similar to ProGen2 [68], we use a 1 token to denote the N-terminal and a 2 token to denote the C-terminal, enabling us to perform both CLM and GLM tasks in the N-to-C or C-to-N direction.

**Position Preserving Fuzzy Encoding** When the infilled residues are transposed to the end of the sequence, they retain their original position ids, and the sentinel span tokens get the same position id as the first token of the infilled span. A drawback of this scheme is that the model will have a strong bias towards exactly the same number of residues that have been removed from the prefix. To mitigate this and allow the model to stop early, for each infilled span, we add a fuzzy length factor to the position ids of all tokens that come after that span. The fuzzy factor is sampled from a geometric distribution with a mean of 0.2 times the length of the span. We provide an example below with infilled spans highlighted in color.

```
Original Sequence: ASVGFKAGVKDYKLTYYTPEYETLDTD
- - - - - - - - - - - - - - - - - - - - - - - - - - - - - - - - - - - - - - - - - - - - -
Infilling Sequence  <bos_glm> 1 A S V G F <span_38>  K  D  Y  K  L  T  Y  Y  T
   Position Ids    :        0      1 2 3 4 5 6      7      12 13 14 15 16 17 18 19 20
<span_6>  L  D  T  D  2 <eos> <span_38> K  A  G  V <eos_span> <span_6> P  E
   21    29 30 31 32 33    34        7      8 9 10 11       12         21    22 23
Y  E  T <eos_span>
24 25 26      27
```

**Ablations**   We validated various design decisions described in the previous paragraphs using validation perplexity, measured on the sequences from the held-out clusters distinct at 50% from the training set for both CLM task (to measure any degradation in CLM performance) and single span infilling. For the latter, we remove a random span of length $L$ from each validation example and measure the perplexity of the suffix after the infilling transformation as described previously (i.e. perplexity of tokens after the `<eos>` token). The results from these ablations can be found in Table 5. Initially, we only infilled spans up to length 10 for infilling and varied the proportion of infilling examples seen during training. We observed that CLM performance shows small degradation (with 1pt perplexity difference) as we vary the proportion from 0 to 1:1, while infilling perplexity varies significantly. We selected a ratio of 1:2 as a compromise. In addition, we also observed that this set of models had very high perplexity for infilling spans of length 50 and 200. Therefore, in the next set of experiments, we vary the length of spans and the mask fraction seen by the model. We see that increasing lengths and mask fraction generally improve performance across all validation sets, validating our choice of using a mixture of Gaussians with mean lengths of up to 400. Finally, we observed that using position-preserving position encoding improved perplexities, while fuzzy length encoding did not lead to any significant performance degradation. Hence, we adopted both of these enhancements.

## B.4   Importance of Warmup Duration

In Section 2.3, we found that the 46B parameter model achieved a validation loss lower than the compute-optimal frontier our scaling laws predicted. A key difference between this model and the smaller ones is that we increased its warmup from 2,000 steps to 10,000 steps to stabilize training (Table 3). This discrepancy could explain the gap in performance, as we find that using 8,000 steps of warmup instead of 2,000 steps improves the final training and validation loss of a 1.4B parameter model trained for 80B tokens (80,000 steps) (Figure 5).

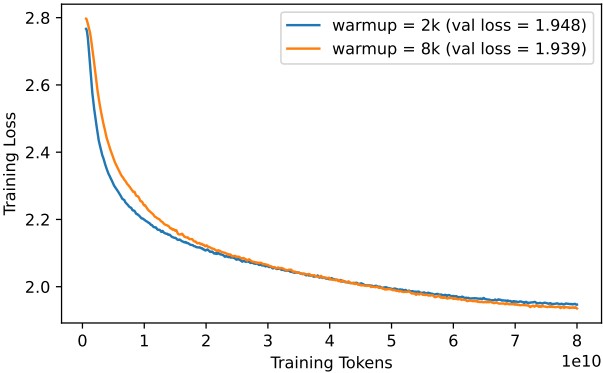

Figure 5: Increasing warmup from 2000 steps to 8000 steps improves the final training and validation loss for a 1.4B parameter model trained on 80B tokens.

| Parameters | Pre-Train Tokens | $\beta$ | $\alpha$ | # Epochs (ProteinGym) | # Epochs (Megascale) |
|---|---|---|---|---|---|
| 339M | 200B | 0.10 | 0.05 | 25 | 6 |
| 1.4B | 500B | 0.10 | 0.05 | 25 | 6 |
| 3B | 500B | 0.10 | 0.02 | 25 | 6 |
| 46B | 1.5T | 0.10 | 0.01 | 25 | 10 |

Table 6: Training hyperparameters for IRPO.

## C Model Alignment

### C.1 Implementation and Hyperparameters

The most relevant alignment algorithms for this paper are direct preference optimization (DPO, [74]) and its generalization iterative reasoning preference optimization (IRPO, [71]). Let $p_\theta$ be the model we are aligning, $p_{\text{ref}}$ be the pre-trained model, $\mathcal{D}$ be our alignment dataset, $x$ be conditioning information, $y \mid x$ be a completion, and $R$ be the reward assigned to $y \mid x$. IRPO can be formulated as

$$\min_\theta \mathbb{E}_{x,((y_1,R_1),...,(y_n,R_n))\sim\mathcal{D}} \sum_{R_i > R_j} -\log\sigma\left(\beta\log\frac{p_\theta(y_i \mid x)}{p_{\text{ref}}(y_i \mid x)} - \beta\log\frac{p_\theta(y_j \mid x)}{p_{\text{ref}}(y_j \mid x)}\right) \quad (3)$$
$$+ \alpha \sum_{R_i=\max R_1,...,R_n} -\frac{\log p_\theta(y_i \mid x)}{|y_i|}.$$

The first term encourages the aligned model $p_\theta$ to assign higher likelihood to preferred completions than dispreferred ones. $\beta$ controls the amount we allow the aligned model $p_\theta$ to deviate from the pre-trained model $p_{\text{ref}}$; larger values allow less deviation. The richness of this loss also increases with the *block size* $n$, i.e. the number of completions that are directly comparable with each other. The second term is the negative log likelihood loss of the completions with the highest reward. If we set the regularization strength $\alpha = 0$, IRPO reduces to DPO.

In this paper, we let the conditioning information $x$ denote a single DMS assay, $y_i$ denote a protein characterized in that assay, and $R_i$ its experimentally measured fitness ($\Delta G$ for the Megascale dataset). We use a cosine learning rate schedule that decays to 50% of the peak learning rate over training and uses the first 10% of steps for a linear warmup. We train all models with peak learning rate $3 \times 10^{-6}$ and peak weight decay $3 \times 10^{-8}$. For models smaller than ProGen3-46B, we run alignment jobs on 8xH100; each job takes at most 3hr. For ProGen3-46B, we require 16xH100, and jobs take 2-6hr, depending on the size of the dataset.

For the ProteinGym experiments, we train a separate set of models for each assay in Table 8. We use a common set of hyperparameters across all configurations; the only exception is that we decrease the batch size from 64 to 32 if the training dataset contains fewer than 1000 samples. Since we are only training on a single assay, the block size $n$ is the same as the batch size.

For all Megascale experiments, we a single model on the entire training set with a batch size of 1024 and block size $n$ of 64. We select the training duration via early stopping on the validation Spearman $\rho$. Table 6 reports additional training hyperparameters that vary between models.

### C.2 IRPO Ablation Studies

We select the $\alpha$ hyperparameter and block size $n$ for each model by training on the Megascale protein stability dataset and evaluating fitness prediction performance on the Megascale validation set, as well as the model's perplexity on the validation set used for pre-training.

Table 7 shows representative trends for a 1.4B parameter model. In general, increasing the block size $n$ improves fitness prediction performance with minimal impact to validation perplexity. This makes sense because the number of comparisons in Equation 3 scales quadratically with $n$, so increasing $n$ imbues the loss with much richer information about the overall fitness landscape. For this reason, we proceed with a fairly large block size of 64 wherever possible.

| Model | $\alpha$ | Block Size | Spearman $\rho$ | Validation Perplexity |
|---|---|---|---|---|
| 1.4B (Pre-Trained) | – | – | 0.471 | 8.150 |
| 1.4B | 0.05 | 8 | 0.562 | 9.384 |
| 1.4B | 0.05 | 16 | 0.639 | 9.607 |
| 1.4B | 0.05 | 64 | 0.643 | 9.712 |
| 1.4B (DPO) | 0 | 64 | 0.670 | 13.874 |
| 1.4B | 0.02 | 64 | 0.650 | 10.551 |
| 1.4B | 0.05 | 64 | 0.643 | 9.712 |
| 1.4B | 0.1 | 64 | 0.629 | 9.241 |

Table 7: Larger block sizes improve fitness prediction performance. Increasing the IRPO regularization coefficient $\alpha$ improves validation perplexity at the cost of fitness prediction performance.

We also find that the NLL loss term of IRPO is an important regularizer that prevents the validation perplexity from degrading too much, i.e. it allows aligned models to retain the broad knowledge about proteins that they gained during pre-training. For each model size, we select the value of $\alpha$ which achieves the best fitness prediction performance while obtaining a validation perplexity no more than 2 points higher than the pre-trained model's. This process yields the values in Table 6.

### C.3 ProteinGym Alignment Experiments

In Figure 6, we report the fitness prediction performance of both pre-trained and aligned models on each of the 8 assays described in Table 8. Figure 4b in the main text reports the average Spearman $\rho$ across these assays. Across the board, aligned models outperform their zero-shot counterparts.

For the five assays with at least 500 single-mutation variants (i.e. the ones where we trained only on single-mutation variants), the performance of the aligned models improves monotonically with their scale. KERMUT and ConFit obtain similar performance to ProGen3-46B on the 4 activity assays; ConFit also performs similarly on CAPSD_AAV2S_Sinai_2021, but KERMUT has meaningfully worse performance on this assay (Table 9).

For the other three assays, there is no such correlation between model scale and fitness prediction performance, though using a larger model almost never causes a meaningful performance drop. This suggests that IRPO learns most effectively from rich single-mutation data, and that on balance, larger models benefit more from alignment than smaller models. Each of the baseline methods also has noticeably worse performance than ProGen3-46B on two of these three assays (Table 9).

We also observe that the two assays where all methods obtain the worst performance (GCN4_YEAST_Staller_2018 and SPG1_STRSG_Wu_2016) both measure binding affinity. Given that the pre-training data contains no signals about a protein's binding partner, predicting a protein's binding affinity is likely a challenging task for models of this sort. However, access to richer single-mutation data may still improve performance for this task.

## D  Split-GFP Methods and Supporting Information

### D.1  Competent cell preparation

Two days in advance, *E. coli* BLl-21 DE3 (either with or without p15a-GFP$_{1\text{-}10}$) was struck onto LB agar with or without kanamycin 50 $\mu$g/L, and incubated overnight at 37°C. One day in advance, a single colony was inoculated into 50 mL LB with or without kanamycin 50 $\mu$g/L in a 250 mL shake flask, and incubated overnight at 37°C 225 RPM. On the day of, the overnight pre-culture was inoculated 1:100 into 500 mL to 2L LB with or without kanamycin 50 $\mu$g/mL in 2L shake flasks, and incubated at 37°C 225 RPM, until OD600 was between 0.4 and 0.6. The flasks were then chilled for 30 minutes on ice and then pelleted by centrifugation (10min, 3500 RCF, 4°C) in chilled 500mL plastic centrifuge bottles. The supernatant was decanted and pellets were suspended in TFB I (100 mM rubidium chloride, 50 mM manganese chloride (MnCl2 H20), 30 mM potassium acetate, 10 mM calcium chloride (CaCl2H2O), 15% v/v glycerol) at a ratio of 1:10 TFBI to starting culture volume. Resuspended cells were incubated on ice for 15 minutes and then pelleted by centrifugation (10min, 3500 RCF, 4°C). The supernatant was decanted and pellets were suspended in TFB II

| Dataset ID | Coarse Selection Type | Max # Mutations (Train) | # Train Variants | Max # Mutations (Test) | # Test Variants |
|---|---|---|---|---|---|
| D7PM05_CLYGR_Somermeyer_2022 | Activity | 1 | 1169 | 23 | 23,346 |
| GFP_AEQVI_Sarkisyan_2016 | Activity | 1 | 1084 | 15 | 50,630 |
| Q6WV12_9MAXI_Somermeyer_2022 | Activity | 1 | 1141 | 13 | 30,260 |
| Q8WTC7_9CNID_Somermeyer_2022 | Activity | 1 | 1201 | 43 | 32,309 |
| GCN4_YEAST_Staller_2018 | Binding | 4 | 617 | 44 | 2021 |
| SPG1_STRSG_Wu_2016 | Binding | 2 | 2167 | 4 | 147,193 |
| CAPSD_AAV2S_Sinai_2021 | Organismal Fitness | 1 | 532 | 28 | 41,796 |
| HIS7_YEAST_Pokusaeva_2019 | Organismal Fitness | 2 | 1643 | 28 | 494,494 |

Table 8: Summary of ProteinGym datasets used for alignment.

| | Avg | D7PM05 | GFP | Q6WV12 | Q8WTC7 | GCN4 | SPG1 | CAPSD | HIS7 |
|---|---|---|---|---|---|---|---|---|---|
| KERMUT | 0.628 | 0.820 | 0.764 | 0.843 | 0.773 | 0.461 | 0.227 | 0.494 | 0.645 |
| ConFit | 0.679 | 0.871 | 0.829 | 0.845 | 0.800 | 0.250 | 0.460 | 0.714 | 0.585 |
| ProGen3-46B | 0.673 | 0.825 | 0.837 | 0.798 | 0.706 | 0.339 | 0.451 | 0.753 | 0.672 |

Table 9: Supervised fitness prediction performance of different methods.

(10mM 3-(N-morpholino)propanesulfonic acid (MOPS), 10mM rubidium chloride, 75 mM calcium chloride, 15% glycerol) at a ratio of 1:25 TFBII to starting culture volume. The resuspensions were incubated on ice for 30 minutes, before being gently homogenized by inversion, and aliquoted to 96w plates, before storage at -80°C.

### D.2 Cloning of expression plasmid backbones

The expression plasmids for the PoI-GFP$_{11}$ and GFP$_{1-10}$, pTet-PoI-GFP$_{11}$ and p15a-GFP$_{1-10}$, respectively, were modeled after Cabantous 2005, synthesized by Twist Bioscience with overlapping fragments, assembled via NEB HiFi following manufacturer recommendations, transformed into 10-beta competent *E. coli* (New England Bioscience), and plated on LB agar containing the appropriate antibiotic. Colonies were picked, miniprepped (Zymo), and sequence verified via long-read nanopore sequencing (Plasmidsaurus) (Table 10).

### D.3 Cloning of the GFP11 fusion libraries

The PoI insert libraries were codon optimized for *E. coli* with DNA Chisel [111], flanked with BsaI Type II restriction sites and synthesized as gene fragments via Twist Bioscience. Plate maps were specified such that generated and wild type proteins from the same cluster were located on the same plate. Positive and negative controls (human dihydrofolate reductase DHFR, NP_000782.1, and human GPCR beta-3 adrenergic receptor, NP_000016.1, respectively) were added to each plate. Additionally, several wells were left empty on each plate to permit background subtraction at the fluorescent measurement stage. DNA fragments were suspended in 100 $\mu$L H20. BsaI-HFv2 Golden Gate reactions were performed using manufacturer recommendations (New England Bioscience) scaled to 10 $\mu$L per assembly. Golden Gate reactions consisted of 30 cycles X (5 min 37°C, 5 min 16°C) followed by 5 min 60°C. 5 $\mu$L completed Golden Gate reaction was transformed following manufacturer recommendations into BL-21 DE3 competent cells either with or without p15a-GFP$_{1-10}$ (New England Bioscience). Transformations were plated onto LB agar plates containing 75 $\mu$g/L spectinomycin either with or without 50 $\mu$g/L kanamycin using a Hamilton liquid handler. Up to four colonies per assembly were picked using a Qpix robotic colony picker into liquid LB containing 75 $\mu$g/L spectinomycin with or without 50 $\mu$g/L kanamycin. Colony PCR was performed using primers splitgpf-PoI-fwd and splitgpf-PoI-rv (Table 11) and RedTaq PCR mastermix (Thermo Fischer) following manufacturer recommendations, scaled to 10 $\mu$L. To spot check, PCR reactions were diluted 1:40 in water and size-verified by capillary electrophoresis (Revvity LabChip).

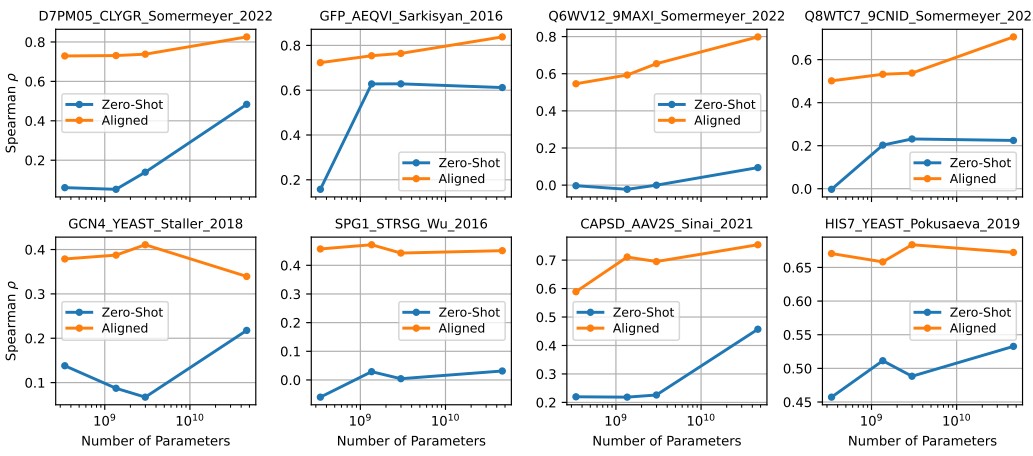

Figure 6: Fitness prediction by zero-shot and aligned models on the assays described in Table 8.

## D.4 Sequence verification

For each plate of assemblies, we generated an in silico pool of references sequences, corresponding to the PCR amplicons using primers splitgpf-PoI-fwd and splitgpf-PoI-rv (Table 11). Prior to sequencing, all pools were pre-validated to ensure sufficient sequence diversity using Jellyfish with kmer size set to 31 [64]. A custom Python script was run across the resulting kmer file to ensure each member of the pool had a kmer with a sufficient edit distance from all other members of the pool.

PCR reactions were pooled, size-selected (1x v:v Beckman Coulter Ampure beads, 2x 80% ethanol wash), and sequenced via Oxford Nanopore PromethION sequencing using R10.4.1 flow cells (Plasmidsaurus). For each sequenced pool, Plasmidsaurus returned a single FASTQ containing all reads. Reference amplicons of the expected library members were compiled into a multi-FASTA. The FASTQ and reference FASTA went through the following analysis steps: All unique 31-mers per reference sequence were identified by Uniquekmer [17]. Reads were filtered based on length. Reads with fewer than 50 bp were discarded using BBDuk [10]. Reads were demultiplexed based on exact match to unique 31-mers using BBSeal [10]. Demultiplexed reads were mapped to reference sequences using minimap2 [58]. Consensus sequence generated based on aligned reads using samtools [25]. Variants were called between the consensus and the reference using DNADiff [54]. Colonies were filtered for sequence fidelity, coverage, and depth (>10 reads covering the ORF), and a single correct representative of each genotype was hitpicked using a Hamilton liquid handler. Hitpick plates were grown overnight in liquid LB containing 75 $\mu$g/L spectinomycin with or without 50 $\mu$g/L kanamycin, mixed 1:1 with 50% glycerol, and stored at -80°C until assay.

## D.5 Split-GFP Assay

In general, the preparation and assay of the PoIs and GFP1-10 and assay followed the protocols delineated in Cabantous, 2006 [11]. The split-GFP assay can be either run in cells using two orthogonal expression plasmids, or in lysate. In a pilot study, we observed a poor correlation between the two methods ($r^2 = -0.27$, $n = 91$ proteins), and that SDS-PAGE abundance was better correlated with GFP fluorescence under sequential induction (Figures 7–9). We further validated the sequential induction approach with SDS-PAGE of 88 proteins (Figure 10). Although the relatively high limit of detection for SDS-PAGE prevents quantitative correlation across the range of measured GFP values, we observed excellent identification of highly expressing and lowly expressing proteins. 19 of 23 proteins in the top quartile had distinct bands at the correct size, while 22 of 22 proteins in the bottom quartile had no detectable band (Figure 8). In the top quartile, two bands were absent. The other two proteins without proper migration were two highly-expressing proteins that failed to migrate on the gel, implying high stability or resistance to SDS-mediated denaturation (Figure 10). While boiling (up to 100°C, 10 min) or proteinase K treatment afforded partial purification, neither

| Plasmid | DNA Sequence 2 |
|---|---|
| pTet-PoI-GFP[11] | ccgccgccctagacctagggcgttcggctgcggcgagcgagcgtatcagctcactcaaaggcggtaatacggttatccacagaatcaggggataacgcaggaaagaacatgtgagcaaaaggccagcaaaaggccaggaaccgtaaaaaggccgcgttg
ctggcgtttttccataggctccgcccccctgacgagcatcacaaaaatcgacgctcaagtcagaggtggcgaaacccgacaggactataaagataccaggcgtttccccctggaagctccctcgtgcgctctcctgttccgaccctgccgcttaccggata
cctgtccgcctttctcccttcggaagcgtggcgctttctcatagctcacgctgtaggtatctcagttcggtgtaggtcgttcgctccaagctgggctgtgtgcacgaacccccgttcagcccgaccgctgcgccttatccggtaactatcgtcttgagtccaa
cccggtaagacacgacttatcgccactggcagcagccactggtaacaggattagcagagcgaggtatgtaggcggtgctacagagttcttgaagtggtggcctaactacggctacactagaagaacagtatttggtatctgcgctctgctgaagccagttaccttc
ggaaaaagagttggtagctcttgatccggcaaacaaaccaccgctggtagcggtggtttttttgtttgcaagcagcagattacgcgcagaaaaaaaggatctcaagaagatcctttgatcttttctacggggtctgacgctcagtggaacgaaaaactca
cgttaagggattttggtcatgactagtgcttggattctcaccaataaaaaacgcccggcggcaaccgagcgcagcgagtcaataatctcgatcaacatcagtatgagttctgactacattctcaacaagagtccaagcttaagaccccacctttcactttagtttttct
aatccgtatatgatcaattcaaggccgaataagaaggctggctctgcaccttggtgatcaaataattcgatagcttgtcgtaatataatggcgcatactatcagtagtaaggttgttttccctttcttctttttaagcagctatgatgctctgatctttcaataacgaacctaaagt
aaaatgccccacagcgctgagtgcataataatgcatctctagtgaaaaacctgttggcataaaaaggctaattgattttcgagagtttcatactgtttttctgtaggccgtgtacctaaatgtacttttgctccatccgacgatgacttagtctgagtacatattctggcctcatgtt
agccttattacgtaaaaaatcttgccagctttccccttcaaagggcaaaagtgagtatggtgcctatctaacatctcaatggctaaggcgtcccgggtatttttatttttacatgccaatacaatgtaggctgctctacaacctagatgctgctctgatcttcaatacatcgaacctaaagt
ggttgttaaacctttcagttccgacctcattaagcagctctaatgcgctgttaatcacttttttatctaatctggacatcattaatgttttattgagctctcgaaccccagagtcccgcattatttgccgactacctgtgatctcgccttttcacgtagtggacaaattc
tccaactgatctgcgcgcgaggccaagcgatcttcttcttgtccaagataagcctgtctagcttcaagtatgaacgggctgatactgggccgttccatggcgttctaccgcggccctccttcaagtgtgtggcaggccgttaccttgccttgccgttactgcgctaccaa
tgcgggacaacgtaagcactacattctgctcatcgccagcccagtcgggcggcgagttccatagcgttaagtttcatttagccgcctcaaatagatcctgttcagaaccggatcaaagagtttcctccgcccgctggacctaccaaggcaaacgctatgttctct
tgctttttgtcagcaagatagccagatcaatgtcgatcgtggctggctcgaagataccgcaagaatgtccattcgcaattccttccctcatgcctgcttagctggataacgccaccgaatgatgtcgtgcgacaacatgtggtctgtgcacaacaatgagcactacctgacgacccac
ggagaaatctcgctctctccaggggaagcgaagttttccaaaaggtcgttgatcaaagctcgccgcgttgtttcatcaagccttacggtcaccgtaaccagcaaatcaatatcactgtgtggcttcaggccgccatccactgcgagccgtacaaatgtacgg
ccagcaacgtcggttcgagatggcgctcgatgacgccaactacctctgatagttgagtcgatacttcggcgatcaacgttctgtctttcttgctgctcacaacctgggttttacctttgttttgggggatcgccctgctgccaatcacgatcaatgtgttaacgta
ggcgtaacgccgttgctgcttttggatgcccgaggcatagactgtacccaaaaaacagtcataacaagccatgaaaaccgccactgcgccgttatccatcgaaactcactcatcatctgtctcttgatcagatctgatccctgcgccatcagatccttggc
ggcaagaaagccatccagtttactttgcagggcttcccaacctaccagagggcgccccagctggcaatccgacgtctaagaaaccattatcatgacattaacctataaaaataggcgtatcacgaggccctttcgtcttcacctcgagtccctatcagtg
atagagattgacatccctatcagtgatagagatactgagcacatcagcaggacgcactgaccgagtcattaaagaggagaaagatacctATGTGAGACCTAATTAATTAATTGGTCTCAGGATCCGATGGAGG
GTCTGGTGGCGGATCAACAAGTCGTGACCACATGGTCCTTCATGAGTACGTAAATGCTGCTGGGATTACATAAGGTACCTAACTCGAGTGAGATCCGGCT
GCTAACAAAGCCCGAAAGGAAGCTGAGTTGGCTGCTGCCACCGCTGAGCAATAACTAGCATAACCTCTAGAGGCCtaacaaataaaacgaaaggctcagtcgaaagactgggc
ctttcgttttatctgttgtttgtcggtgaacgctctcctgagtaggacaaat |
| p15a-GFP[1-10] | TTTCCAGTCGGGAAACCTGTCGTGCCAGCTGCATTAATGAATCGGCCAACGCGCGGGGAGAGGCGGTTGCGTATTGGGCGCCAGGGTGGTTTTTCTTTT
CACCAGTGAGACGGGCAACAGCTGATTGCCCTTCACCGCTGGCCCTGAGAGAGTTGCAGCAAGCGGTCCACGCTGGTTTGCCCCAGCAGGCGAAAATC
CTGTTTGATGGTGGTTAACGGCGGGATATAACATGAGCTGTCTTCGGTATCGTCGTATCCCACTACCGAGATATCCGCACCAACGCGCAGCCCGGACTCG
GTAATGGCGCGCATTGCGCCCAGCGCCATCTGATCGTTGGCAACCAGCATCGCAGTGGGAACGATGCCCTCATTCAGCATTTGCATGGTTTGTTGAAAAC
CGGACATGGCACTCCAGTCGCCTTCCCGTTCCGCTATCGGCTGAATTTGATTGCGAGTGAGATATTTATGCCAGCCAGCCAGACGCAGACTGTAACGCGCCTAGCGACAC
AGAACTTAATGGGGCCCGCTAACAGCGCGATTTGCTGGTGACCCAATGCGACCAGATGCTCCACGCCCAGTCGCGTACCGTCTTCATGGGAGAAAATAAATA
CTGTTGATGGGTGTCTGGTCAGAGACATCAAGAAATAACGCCGGAACATTAGTGCAGGCAGCTTCCACAGCAATGGCATCCTGGTCATCCAGCGGATAG
TTAATGATCAGCCCACTGACGCGTTGCGCGAGAAGATTGTGCACCGCCGCTTTACAGGCTTCGACGCCGCTTCGTTCTACCATCGACACCACCACGCTGG
CACCCAGTTGATCGGCGCGAGATTTAATCGCCGCGACAATTTGCGACGGCGCGTGCAGGGCCAGACTGGAGGTGGCAACGCCAATCAGCAACGACTGTT
TGCCCGCCAGTTGTTGTGCCACGCGGTTGGGAATGTAATTCAGCTCCGCCATCGCCGCTTCCACTTTTTCCCGCGTTTTCGCAGAAACGTGGCTGGCCTGG
TTCACCACGCGGGAAACGGTCTGATAAGAGACACCGGCATACTCTGCGACATCGTATAACGTTACTGGTTTCACATTCACCACCCTGAATTGACTCTCTTC
CGGGCGCTATCATGCCATACCGCGAAAGGTTTTGCGCCATTCGATGGTGTCCGGGATCTCGACGCTCTCCCTTATGCGACTCCTGCATTAGGAAGCAGCC
CAGTAGTAGGTTGAGGCCGTTGAGCACCGCCGCCGCAAGGAATGGTGCATGCAAGGAGATGGCGCCCAACAGTCCCCGGCCACGGGGCCTGCCACCAT
ACCCACGCGAAACAAGCGCTCATGAGCCCGAAGTGGCGAGCCCGATCTTCCCCATCGGTGATGTCGGCGATATAGGCGCCAGCAACCGCACCTGTGGC
GCCGGTGATGCCGGCCACGATGCGTCCGGCGTAGAGGATCGAGATCTCGATCCCGCGAAATTAATACGACTCACTATAGGGGAATTGTGAGCGGATAAC
AATTCCCCTCTAGGATCCGAATTCGAGCTCTCTAGAGAAAGAGGAGAAATACTAGATGAGCAAAGGAGAAGAACTTTTCACTGGAGTTGTCCCAATTCTT
GTTGAATTAGATGGTGATGTTAATGGGCACAAATTTTCTGTCAGAGGAGAGGGTGAAGGTGATGCTACAATCGGAAAACTCACCCTTAAATTTATTTGCA
CTACTGGAAAACTACCTGTTCCATGGCCAACACTTGTCACTACTCTGACCTATGGTGTTCAATGCTTTTCCCGTTATCCGGATCACATGAAAAGGCATGAC
TTTTTCAAGAGTGCCATGCCCGAAGGTTATGTACAGGAACGCACTATATCTTTCAAAGATGACGGGAAATACAAGACGCGTGCTGTAGTCAAGTTTGAAG
GTGATACCCTTGTTAATCGTATCGAGTTAAAGGGTACTGATTTTAAAGAAGATGGAAACATTCTCGGACACAAACTCGAGTACAACTTTAACTCACACAA
TGTATACATCACGGCAGACAAACAAAAGAATGGAATCAAAGCTAACTTCAACAGTTCGCCACAACGTTGAAGATGGTTCCGTTCAACTAGCAGACCATTAT
CAACAAAATACTCCAATTGGCGATGGCCCTGTCCTTTTACCAGACAACCATTACCTGTCGACACAAACTGTCCTTTCGAAAGATCCCAACGAAAAGTAAA
AGCTTGCGGCCGCACTCGAGCACCACCACCACCACCACTGAGATCCGGCTGCTAACAAAGCCCGAAAGGAAGCTGAGTTGGCTGCTGCCACCGCTGAGC
AATAACTAGCATAACCTCTTGGGGCCTCTAAACGGGTCTTGAGGGGTTTTTGCTGAAAGGAGGAACTATATCCGGATTGGCGAATGGGAGCTGCCCCTGT
AGCGGCGCATTAAGCGCGGCGGGTGTGGTGGTTACGCGCAGCGTGACCGCTACACTTGCCAGCGCCCTAGCGCCCGCTCCTTTCGCTTTCTTCCCTTCCTT
TCTCGCCACGTTCGCCGGCTTTCCCCGTCAAGCTCTAAATCGGGGCTCCCTTTAGGGTTCCGATTTAGTGCTTTACGGCACCTCGACCCCAAAAAACTTG
ATTAGGGTGATGGTTCACGTAGTGGGCCATCGCCCTGATAGACGGTTTTTCGCCCTTTGACGTTGGAGTCCACGTTCTTTAATAGTGGACTCTTGTTCCAA
ACTGGAACAACACTCAACCCTATCTCGGTCTATTCTTTTGATTTATAAGGGATTTTGCCGATTTCGGCCTATTGGTTAAAAAATGAGCTGATTTAACAAAA
ATTTAACGCGAATTTTAACAAAATATTAACGCTTACAATTTAGGTGGCACTTTTCGGGGAAATGTGCGCGGAACCCCTATTTGTTTATTTTTCTAAATACA
TTCAAATATGTATCCGCTCATGAATTAATTCTTAGAAAAACTCATCGAGCATCAAATGAAACTGCAATTTATTCATATCAGGATTATCAATACCATATTTT
TGAAAAAGCCGTTTCTGTAATGAAGGAGAAAACTCACCGAGGCAGTTCCATAGGATGGCAAGATCCTGGTATCGGTCTGCGATTCCGACTCGTCCAACAT
CAATACAACCTATTAATTTCCCCTCGTCAAAAATAAGGTTATCAAGTGAGAAATCACCATGAGTGACGACTGAATCCGGTGAGAATGGCAAAAGTTTATG
CATTTCTTTCCAGACTTGTTCAACAGGCCAGCCATTACGCTCGTCATCAAAATCACTCGCATCAACCAAACCGTTATTCATTCGTGATTGCGCCTGAGCGA
GACGAAATACGCGATCGCTGTTAAAAGGACAATTACAAACAGGAATCGAATGCAACCGGCGCAGGAACACTGCCAGCGCATCAACAATATTTTCACCTG
AATCAGGATATTCTTCTAATACCTGGAATGCTGTTTTCCCGGGGATCGCAGTGGTGAGTAACCATGCATCATCAGGAGTACGGATAAAATGCTTGATGGT
CGGAAGAGGCATAAATTCCGTCAGCCAGTTTAGTCTGACCATCTCATCTGTAACATCATTGGCAACGCTACCTTTGCCATGTTTCAGAAACAACTCTGGCG
CATCGGGCTTCCCATACAATCGATAGATTGTCGCACCTGATTGCCCGACATTATCGCGAGCCCATTTATACCCATATAAATCAGCATCCATGTTGGAATTT
AATCGCGGCCTAGAGCAAGACGTTTCCCGTTGAATATGGCTCATAACACCCCTTGTATTACTGTTTATGTAAGCAGACAGTTTTATTGTTTGcgcacaacttatatcg
tatggggctgacttcaggtgctacattgaagagataaattgcactgaaatctagaaatattttatctgattaataagatgatcttcttgagacgtttttggtctgcgcgtaatctcttgctctgaaaacgaaaaaccgccttgcagggcggtttttcgaaggttcct
gagctaccaactctttgaaccgaggtaactggcttggaggagcgcagtcaccaaaacttgtccttttcagtttagccttaaccggcgcatgacttcaagactcaagctaactcctctaaatcaattaccagtggctgctgccagtggtgcttttgcatgtctttccgggttgg
actcaagacgatagttaccggataaggcgcagcggtcgggctgaacggggggttcgtgcatacagtccagcttggagcgaactgacctgaaagcgtggaatgagacaaacgcggccataacagcgataagtccgcctacaccgaactgagatacctacagcgtgagc
cgaaaggcaggaacaggagagcgcacgagggagccgccaggggaaacgcctggtatctttatagtcctgtcgggtttcgccaccactgatttgagcgtcagatttcgtgatgcttgtcagggggcggagcctatggaaaaacgccttgccgccggc
cctctcacttccctgttaagtctcgcctggttgtatttttcctatcctgcagggaagctgcgattctcatctcatatatatgccaatatatctccccgccctaaatatactccgctagccgactacatactatcctgatcacatattctgctgacgcacccggcc
agcctttttctctgccacatgaagcacttcactgacccagcccgtcccatcagtgcaaataactatcccgctagccgactacatactgctgacacgcactatgcTTGCTCAGGTCGCAGACGTTTTGCAGCAGCAGTCGCTTCACGTTC
GCTCGCGTATCGGTGATTCATTCTGCTAACCAGTAAGGCAACCCCGCCAGCCTAGCCGGGTCCTCAACGACAGGAGCACGATCATCGTCGCACCCGTGGGGC
CGCCATGCCGGCGATAATGGCCTGCTTCTCGCCGAAACGTTTGGTGGCGGGACCAGTGACGAAGGCTTGAGCGAGGGCGTGCAAGATTCCGAATTACCGC
AAGCGACAGGCCGATCATCGTCGCGCTCCAGCGAAAGCGGTCCTCGCCGAAAATGACCCAGAGCGCTGCCGGCACCCTGTCCTACGAGTTGCATGATAAA
GAAGACAGTCATAAGTGCGGCGACGATAGTCATGCCCCGCGCCCACCGGAAGGAGCTGACTGGGTTGAAGGCTCTCAAGGGCATCGGTCGAGATCCCGG
TGCCTAATGAGTGAGCTAACTTACATTAATTGCGTTGCGCTCACTGCCCGC |

Table 10: Plasmids used in this study.

| Plasmid | DNA Sequence 2 |
|---|---|
| splitgpf-PoI-fwd | gcaggacgcactgaccgagttc |
| splitgpf-PoI-rv | agggcggcggatttgtcctact |

Table 11: Primers used for the size and sequence validation of PoI inserts in pTet-PoI-GFP[11].

led to proper migration of the bands. Interestingly, both of these proteins shared the same cluster with the most closely related natural protein 3-hydroxyacyl-[acyl-carrier-protein] dehydratase FabZ.

Due to the higher correlation with SDS-PAGE, we chose the in-lysate approach for experimental validation of the broader set. We describe both approaches in more detail below.

### D.5.1 Whole-cell assay

Glycerol stocks containing pTet-PoI-GFP[11] and p15a-GFP[1-10] plasmids were inoculated into 1 mL liquid LB containing 75 $\mu$g/L spectinomycin and 50 $\mu$g/liter kanamycin sulfate and grown overnight in a shaking plate incubator (37°C, 1000RPM). In the morning, overnight growth was inoculated in quadruplicate plates 1:100 in LB containing 75 $\mu$g/L spectinomycin and 50 $\mu$g/liter kanamycin sulfate, and grown in a shaking plate incubator (37°C, 1000RPM) until OD600 reached 0.5 to 1.0. PoI-GFP[11] expression was induced by the addition of anhydrotetracycline in ethanol to a final concentration of 250 ng/L. Simultaneous GFP[1-10] was induced by the addition of isopropyl $\beta$-D-

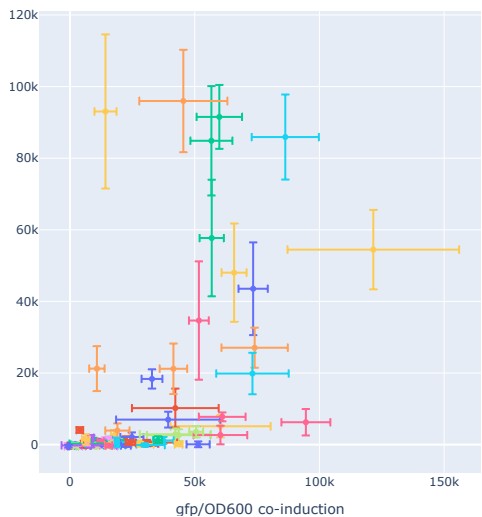

Figure 7: Scatter plot of mean split-GFP under sequential induction (y-axis) vs mean co-induction (x-axis), t = 16 hours, for 91 unique proteins. Each color is a unique protein sequence. Error bars are standard deviation of n = 4 replicates.

thiogalactopyranoside (IPTG) to a final concentration of 0.1 mM. Protein production continued for two hours in a shaking incubator (37°C, 1000RPM). Cultures were diluted 1:1 in LB to a final volume of 100 $\mu$L in a 384-well black clear bottom plate (Greiner 781097). Fluorescence and OD600 were read on a Tecan Spark plate reader (Ex: 475nm, Em: 530 nm, Gain: 69, Z-position: 30000).

### D.5.2 Lysate assay

**Preparation of protein of interest:** Glycerol stocks containing pTet-PoI-GFP11 plasmids were inoculated into 1mL liquid LB containing 75 $\mu$g/L spectinomycin and grown overnight in a shaking plate incubator (37°C, 1000RPM). In the morning, overnight growth was inoculated 1:50 in quadruplicate in 1mL LB containing 75 $\mu$g/L spectinomycin, and grown in a shaking plate incubator (37°C, 1000RPM) until OD600 reached 0.5 to 1.0. PoI was induced by the addition of anhydrotetracycline in ethanol to a final concentration of 250 ng/L. Protein production continued for two hours in a shaking incubator (37°C, 1000RPM). Culture-containing plates were spun down (3500 RCF, 5 minutes), decanted, and 100 $\mu$L BugBuster MasterMix (EMD Millipore) containing 25 $\mu$g/mL chloramphenicol was added to lyse cells and stop protein expression. Cells were lysed by incubation in a shaking incubator (25°C, 300RPM, 1 hour). Lysates were stored overnight at 4° before assay.

**Preparation of GFP1-10** p15a-GFP$_{1-10}$ was transformed into *E. coli* BL-21 DE3 competent cells (New England Bioscience) following manufacturer recommendations. A single colony was grown overnight in LB 50 $\mu$g/liter kanamycin sulfate, diluted 1:1 with 50% glycerol, and stored at -80°C. Biomass from the glycerol stock was inoculated into 50 mL LB 50 $\mu$g/liter kanamycin sulfate in a 250 mL shake flask, and incubated overnight (37°C, 225 RPM). The following morning, the overnight culture was inoculated 1:100 into LB 50 $\mu$g/liter kanamycin sulfate in a 2 L shake flask. After two hours (37°C, 225 RPM), GFP$_{1-10}$ expression was induced with a final concentration of 1 mM isopropyl $\beta$-D-thiogalactopyranoside (IPTG). After five hours of incubation (37°C, 225 RPM), the culture was placed at 4° and left overnight. The following morning, the culture was spun down (3500 RCF x 15 minutes) and re-suspended in 1/5 of the culture volume (i.e. 200 mL for 1 L culture) in lysis buffer containing 100 mm Tris HCl Ph 7.5, 10% glycerol, and 150 mm sodium chloride (TNG buffer). The resuspended culture was sonicated on ice (VWR 76193-590 50% duty, 10 minutes, 30 second pulse). The lysate was centrifuged (3500 RCF x 15 minutes) and the GFP$_{1-10}$-containing supernatant was decanted.

### D.5.3 In trans folding and fluorescence measurement

10 $\mu$L of PoI-$_{\text{GFP11}}$ lysate were added to 90uL of the GFP$_{1\text{-}10}$ lysate in a 384-well black clear bottom plate (Greiner 781097). Plates were sealed and incubated at 25°C overnight. Fluorescence was read on a Tecan Spark plate reader (Ex: 475nm, Em: 530 nm, Gain: 69, Z-position 30000). Samples were background subtracted using the mean fluorescence of the empty wells on a per plate basis.

### D.6 SDS-PAGE validation

10 $\mu$L of cell lysate were diluted with 30 $\mu$L BugBuster Protein Extraction Reagent with 25 $\mu$g/mL chloramphenicol and then added to 12 $\mu$L Laemmli + DTT and pipet mixed. Samples were heated at 85°C for 3 min. 10 $\mu$L were loaded on a Mini-PROTEANő TGX Precast Gel (BioRad) and run at 120V for 50 minutes. Gels were rinsed in DI water then opened and transferred into troughs filled with DI water. Troughs were gently shaken for 15 minutes on an Ohaus light duty orbital shaker at 130 rpm at room temperature. Then DI water was replaced and this wash process was repeated two more times. Gels were then stained with GelCode Blue Stain Reagent for 45 minutes while shaking at 130 rpm. Gel was de-stained by decanting dye and adding DI water and shaking at 130 rpm for another hour. Photographs were taken via smartphone camera.

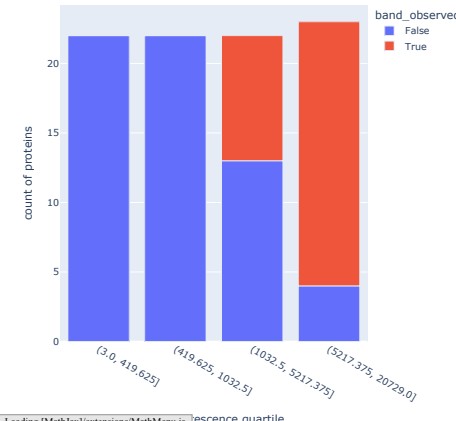

Figure 8: Count of SDS-PAGE bands observed at correct size for 92 proteins, organized by GFP quartile.

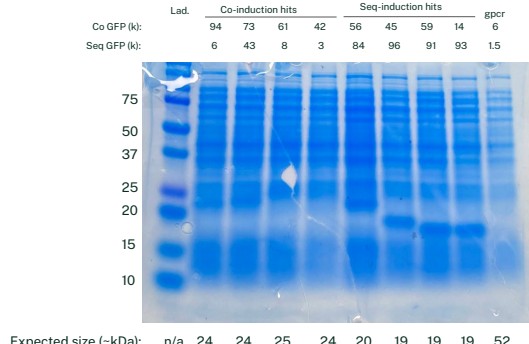

Figure 9: SDS-PAGE of 9 proteins. Top rows show mean GFP fluorescence in either co- or sequential induction experiments, t = 16 hours. The bottom row shows expected size in kDa.

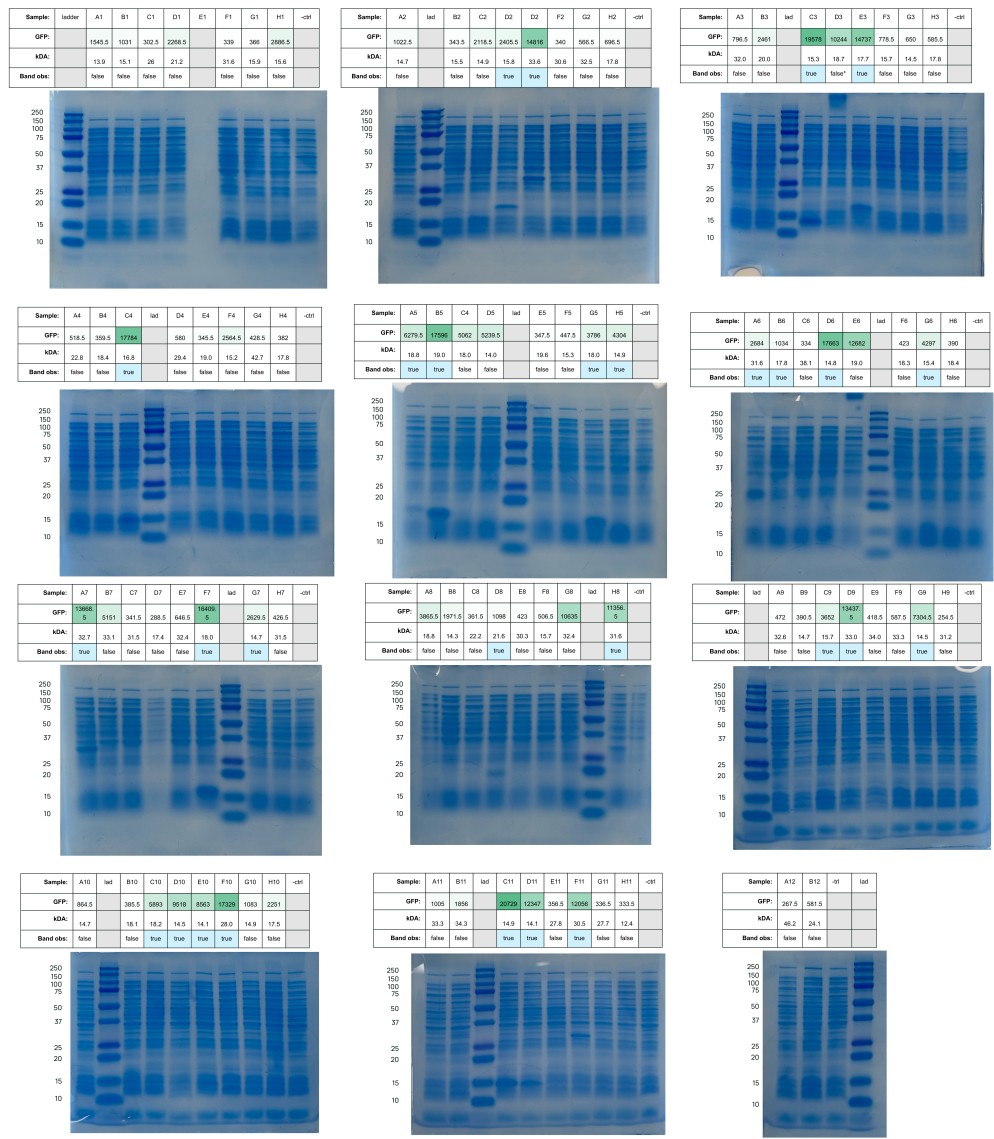

Figure 10: SDS-PAGE gels of induced proteins. Each lane is labeled by plate position, background-subtracted GFP, and boolean call on band presence at the anticipated size. Negative control is uninduced plasmid.

