# OpenReview forum: "Scaling Unlocks Broader Generation and Deeper Functional Understanding of Proteins"
_NeurIPS.cc/2025/Conference — NeurIPS 2025 spotlight_

### Official Review · Reviewer_Hphy · 2025-07-01

**Clarity:** 4
**Significance:** 3
**Originality:** 4
**Rating:** 5
**Confidence:** 4

**Summary:**

This paper presents a systematic study of scaling laws in protein language models (PLMs), using a newly curated dataset called PPA that contains over 150 million protein sequences grouped into clusters at different sequence identity thresholds. The authors train a suite of ProGen3 models ranging from 44M to 4.7B parameters, analyzing how model scale and training data distribution affect generative quality, functional diversity, and downstream performance.

**Questions:**

- Is there a particular reason why Figures 1e and 4a only show a single FLOP-performance point for the 4B model, whereas other model sizes are represented with full scaling curves?

- The paper mentions that segments were randomly masked for the infilling objective. Could this result in overly easy infilling tasks compared to biologically meaningful challenges? Please consider discussing alternative infilling objectives, such as fixed-region scaffolding or motif-conditioned design, which may provide more biologically relevant evaluation.

- While the paper focuses on scaling within the ProGen3 framework, incorporating additional baseline models such as ESM-3 into perplexity, diversity, or cluster coverage plots (even if just as reference lines) would better contextualize ProGen3’s performance relative to existing methods.

**Ethical Concerns:**

["NO or VERY MINOR ethics concerns only"]

**Final Justification:**

The authors have addressed my questions/concerns. I would keep my recommendation for the acceptance of this submission.

**Limitations:**

yes

**Quality:**

4

**Strengths And Weaknesses:**

Strengths
- The paper introduces a well-curated and biologically meaningful dataset (PPA), organized with sequence clustering and rigorous data splits, which provides a strong foundation for benchmarking PLMs.

- It presents a thorough empirical study of how training distribution and model scale influence protein generation, offering insights that can benefit the field without requiring similarly expensive experiments.nd similar computational cost, especially in academic settings, where resources are constrained.

- Beyond language modeling metrics like perplexity, the study evaluates sequence diversity and functional viability using in vitro experiments on generated sequences from the luciferase family. The paper offers practical insights into how data and model scaling impact protein generation and design, particularly in resource-constrained academic settings.


Weakness
- While the paper makes strong contributions in dataset curation, model scaling, and evaluation, it would benefit from a clear commitment to open-sourcing resources. In particular, the authors should clarify whether they intend to release the PPA dataset, trained model checkpoints, and training/inference code. Making these resources public would significantly enhance the reproducibility and long-term impact of the work.

- While ProGen3 demonstrates promising performance, virtually all benchmarks presented in the paper are internal—comparing different variants of ProGen3 across model sizes or training distributions. Since this work introduces a new model family, it would be valuable to include direct comparisons with existing models such as ESM-3 to better contextualize ProGen3’s capabilities relative to the current state of the art.

---

> ### Author Rebuttal · Authors · 2025-07-31
>
> Thank you for your thoughtful feedback. We want to briefly correct the claim that “The authors train a suite of ProGen3 models ranging from 44M to 4.7B parameters” -- ProGen3 models range in size from 112M to 46B parameters. Addressing your points one at a time,
> 1. We have open sourced all model sizes up to 3B parameters, but we do not intend to open source PPA-1 or ProGen3-46B.
> 2. In Section 4, we do compare applying IRPO to ProGen3 against state-of-the-art supervised fitness prediction methods ProteinDPO [1] for stability prediction, and ConFit [2] and KERMUT [3] for general fitness prediction. However, we will add additional comparisons to ProGen2-XL [4], a 6.4B parameter dense autoregressive model, for both unconditional generation and alignment. We find that only only 4.1% of generations from ProGen2-XL (out of 4M total) are valid proteins (compared to 39.8% / 49.3% from ProGen3-3B/46B), and they cover 0.82x / 0.66x as many clusters as ProGen3-3B/46B. For a fixed parameter count, ProGen3 also outperforms ProGen2 on all our alignment benchmarks, but ProGen3 is much more efficient due to its sparse MoE implementation. We do not compare against ESM3 [5] because its license precludes us from using it, and we do not compare against AIDO.Protein [6] or ESM-IF1 [7] because they perform structure-conditioned generation, not unconditional generation.
> 3. Figures 1e and 4a only show a single point for 46B parameters because we only trained this model size on 1.5T tokens. Because we need to train a separate model for each dataset size (Lines 117-118), it was prohibitively expensive to train multiple versions of the 46B parameter model. Instead, we used our scaling laws to determine the right parameter/token split for 1.1e23 FLOPs.
> 4. We do not expect that masking random spans would yield an overly easy infilling objective. First, we still apply autoregressive attention to the full sequence, so infilling a span of length L is just as hard as generating the L residues at the C-terminal when performing N-to-C generation. Second, we train to infill long spans that can cover up to 80% of the protein and can be >400 residues (Lines 856-862). That said, your suggestions to apply fixed-region scaffolding or motif-conditioned design would be interesting extensions that could improve performance in fine-tuning regimes where more structural data is available; unfortunately, they are infeasible to apply on large-scale unlabeled datasets like PPA-1.
>
> [1] Widatalla et al. “Aligning protein generative models with experimental fitness via Direct Preference Optimization.” bioRxiv, 2024.
>
> [2] Zhao et al. “Contrastive Fitness Learning: Reprogramming Protein Language Models for Low-N Learning of Protein Fitness Landscape.” RECOMB, 2024.
>
> [3] Groth et al. “Kermut: Composite kernel regression for protein variant effects.” NeurIPS, 2024.
>
> [4] Nijkamp et al. “ProGen2: Exploring the boundaries of protein language models.” Cell Systems, 2023.
>
> [5] Hayes et al. “Simulating 500 million years of evolution with a language model.” Science, 2025.
>
> [6] Sun et al. “Mixture of Experts Enable Efficient and Effective Protein Understanding and Design.” bioRxiv, 2024.
>
> [7] Hsu et al. "Learning inverse folding from millions of predicted structures." PMLR, 2022.

---

### Official Review · Reviewer_RM9H · 2025-07-02

**Clarity:** 3
**Significance:** 2
**Originality:** 3
**Rating:** 4
**Confidence:** 4

**Summary:**

This paper presents ProGen3, a sparse mixture-of-experts generative protein language model. The authors explore Scaling-Laws in this architecture and introduce an optimized pre-training dataset, PPA v1. Experimental results demonstrate that larger PLMs can generate much wider diversity across protein families. Furthermore, after using iterative reasoning preference optimization (IRPO) with wet lab data, ProGen3 demonstrates strong performance on protein fitness prediction tasks.

**Questions:**

1. The authors show that IRPO with wet lab data improves ProGen3’s performance on protein fitness prediction. Does this strategy generalize to other downstream tasks (e.g., protein property prediction), or would each task require different experiments? Additional results on various tasks would strengthen the value of wet lab results.
2. The authors demonstrate the effectiveness of IRPO with wet lab data. Is this approach model-agnostic? If the same data and IRPO method were applied to ProteinDPO [5] or other PLMs, could even better results be obtained? Such discussion or experiments would strengthen the paper’s contribution to data and experimental methodology.



[1] Simulating 500 million years of evolution with a language model

[2] Mixture of Experts Enable Efficient and Effective Protein Understanding and Design

[3] ProGen2: Exploring the boundaries of protein language models

[4] Training Compute-Optimal Protein Language Models

[5] Aligning protein generative models with experimental fitness via Direct Preference Optimization

**Ethical Concerns:**

["NO or VERY MINOR ethics concerns only"]

**Final Justification:**

The authors' reply solve my questions. I decide to raise my rating.

**Limitations:**

yes

**Quality:**

3

**Strengths And Weaknesses:**

Strengths:

1. The authors curate and present the PPA-1 dataset, showing the advantages of training on de-biased data.
2. They optimize the LLM with wet lab data and highlight the effectiveness of such experimental reinforcement.
3. They train a large (46B parameter) generative protein language model, providing a new base model for the field.

Weaknesses:

1. The paper lacks comparisons of protein generation capabilities with other PLM models, such as ESM3 [1], AIDO.Protein [2], and ProGen2 [3], all of which also support protein generation. It would strengthen the work to include direct comparisons to demonstrate ProGen3’s advantages over these models.
2. In Section 3.1, the authors claim that “larger PLMs generate more diverse proteins.” However, larger models were trained with more data (Lines 135-137), so it is unclear if the gains come from the increased model capacity or simply from more diverse training data. The necessity of scaling up model size should be better demonstrated.
3. Methodologically, the innovations seem somewhat incremental. Prior works have already validated MoE architectures for PLMs [2], explored scaling laws with different PLM architectures [3], and shown that preference optimization improves model fitness prediction ability [5]. While the value of using wet lab data is acknowledged, the technical novelty warrants further clarification.

---

> ### Author Rebuttal · Authors · 2025-07-31
>
> Thank you for your thoughtful feedback. Addressing your points one at a time,
> 1. In Section 4, we do compare applying IRPO to ProGen3 against state-of-the-art supervised fitness prediction methods ProteinDPO [1] for stability prediction, and ConFit [2] and KERMUT [3] for general fitness prediction. However, we will add additional comparisons to ProGen2-XL [4], a 6.4B parameter dense autoregressive model, for both unconditional generation and alignment. We find that only only 4.1% of generations from ProGen2-XL (out of 4M total) are valid proteins (compared to 39.8% / 49.3% from ProGen3-3B/46B), and they cover 0.82x / 0.66x as many clusters as ProGen3-3B/46B. For a fixed parameter count, ProGen3 also outperforms ProGen2 on all our alignment benchmarks, but ProGen3 is much more efficient due to its sparse MoE implementation. We do not compare against ESM3 [5] because its license precludes us from using it, and we do not compare against AIDO.Protein [6] or ESM-IF1 [7] because they perform structure-conditioned generation, not unconditional generation.
> 2. While it is true that the larger models in Section 3 were trained on more data, the point still stands that generation diversity improves when using more pre-training FLOPs. The mechanism is simple: a lower validation loss implies a better understanding of the natural protein distribution, especially for rare or out-of-distribution proteins (Lines 93-97). However, to address your concern more directly, we will create supplementary figures that compare the quality and diversity of generations from ProGen3-339M trained on 200B tokens (PG3-339M_200B), ProGen3-3B trained on 200B tokens (PG3-3B_200B), and ProGen3-3B trained on 500B tokens (PG3-3B_500B). We find that increasing ProGen3-3B’s dataset size from 200B to 500B tokens has a much smaller impact than scaling up from 339M to 3B parameters. 30.0%, 37.8%, and 39.8% of the generations are valid from PG3-339M_200B, PG3B-3B_200B, and PG3-3B_500B, respectively. Additionally, PG3-3B_500B covers 81% more clusters than PG3-339M_200B, but only 14% more clusters than PG3-3B_200B.
> 3. We are the first to systematically study the effect of the pre-training data distribution for PLMs, and we find that it has a significant effect on model performance (Figure 1d). Given that prior work has shown that the data distribution [8] and pre-training task [9] can affect scaling behavior, we feel that deriving compute-optimal scaling laws for this optimized distribution and for the infilling task are significant contributions. The fact that our results agree with Cheng et al. [9], who operate in a totally different setting, demonstrates the robustness of these scaling laws for autoregressive PLMs more broadly. We also performed the first wet lab experiment to characterize the impact of model scaling on the quality and diversity of generated sequences, and the findings in Section 3 are highly novel. Finally, while prior work has shown that preference optimization with wet lab data improves PLM performance, we are the first to show that larger pre-trained models yield better aligned models, especially in few-shot settings (Figure 4b).
> 4. IRPO can be used to increase the correlation of a model’s likelihood with any metric that we wish to optimize, but it is not suited for multi-class classification tasks. Section 4 shows that IRPO can greatly improve the correlation of ProGen3’s likelihoods with a wide range of protein properties. For general properties like thermostability, we show that aligned models can generalize to protein families that are highly distinct from those they were trained on, both for fitness prediction and sequence generation (Lines 251-254, Figures 4c-4g). We also show that for properties that are more specific to a protein family, including activity, binding affinity, and organismal fitness, as few as 500 experimental samples (a modest number that many labs can reasonably generate) can greatly improve the model’s performance for that specific task (Figure 4b, Supplementary Tables 8-9).
> 5. IRPO is a model-agnostic method. It is a variant of DPO that adds a negative log likelihood term to DPO’s contrastive loss (Appendix D.1). To strengthen our paper and address your concerns about having an apples-to-apples comparison with a different model family, we will include the results of applying IRPO to ProGen2-XL [4] in our revisions (see point 1 for more discussion of baselines). As mentioned above, we find that ProGen3 outperforms ProGen2 across the board. That said, similar to how IRPO is a variant of DPO that we found was well-optimized for ProGen3, ProteinDPO [1] is a variant of DPO optimized for stability prediction with ESM-IF1 [7], an inverse folding model that generates a protein sequence conditioned on its structure, and ConFit [2] is a variant of DPO that is optimized for ESM2 [10], a masked language model. Note that since masked language models don’t have well-defined likelihoods, vanilla DPO/IRPO cannot be applied to them. Moreover, we used identical train/val/test splits to evaluate each method. Thus, the baselines already in our paper are essentially the comparisons you are asking for, and we show that the aligned ProGen3-46B outperforms ProteinDPO and matches ConFit while being a much more flexible model that is capable of unconditional sequence generation.
>
> [1] Widatalla et al. “Aligning protein generative models with experimental fitness via Direct Preference Optimization.” bioRxiv, 2024.
>
> [2] Zhao et al. “Contrastive Fitness Learning: Reprogramming Protein Language Models for Low-N Learning of Protein Fitness Landscape.” RECOMB, 2024.
>
> [3] Groth et al. “Kermut: Composite kernel regression for protein variant effects.” NeurIPS, 2024.
>
> [4] Nijkamp et al. “ProGen2: Exploring the boundaries of protein language models.” Cell Systems, 2023.
>
> [5] Hayes et al. “Simulating 500 million years of evolution with a language model.” Science, 2025.
>
> [6] Sun et al. “Mixture of Experts Enable Efficient and Effective Protein Understanding and Design.” bioRxiv, 2024.
>
> [7] Hsu et al. "Learning inverse folding from millions of predicted structures." PMLR, 2022.
>
> [8] Mayilvahanan et al. “LLMs on the Line: Data Determines Loss-to-Loss Scaling Laws.” ICML, 2025.
>
> [9] Cheng et al. “Training Compute-Optimal Protein Language Models.” NeurIPS, 2024.
>
> [10] Lin et al. “Evolutionary-scale prediction of atomic-level protein structure with a language model.” Science, 2023.

---

### Official Review · Reviewer_RnRR · 2025-07-02

**Clarity:** 4
**Significance:** 4
**Originality:** 3
**Rating:** 5
**Confidence:** 4

**Summary:**

This paper presents a suite of MoE PLMs named ProGen3 that demonstrate strong scaling performance on not only the pretraining task, but also after being finetuned for downstream tasks such as fitness prediction and stable protein sequence generation. Wet lab experiments also verify that the generated sequences are diverse and have high expression rate.

**Questions:**

1. Would the fitness prediction performance improve if you used an alignment algorithm that incorporated the actual experimental fitness values (e.g. offline/off-policy RLHF or policy gradient), rather than only ranked pairs?

**Ethical Concerns:**

["NO or VERY MINOR ethics concerns only"]

**Final Justification:**

I remain satisfied with the paper and did not identify any reasons to either increase or decrease my score after the discussion.

**Limitations:**

yes

**Quality:**

4

**Strengths And Weaknesses:**

Strengths:
- The authors conduct an extensive scaling study on the output proteins (and not just the embeddings) of ProGen3 at multiple model sizes and with a validation metric that also accounts for OOD proteins. Such scaling studies are very underexplored in the PLM literature. The authors also empirically identified a data mixture that yielded lower OOD loss than a uniform or unmodified mixture.
- A large proportion of the generated proteins have are viable/expressible, even for the generations with <30%ID to any of the natural proteins in the data.
- The ProGen3 models also demonstrate strong performance when finetuned for downstream tasks such as fitness prediction and generation of stable proteins.
- The ProGen3 suite of models is also provided at a wide range of model scales, making it suitable for a variety of downstream uses as well as further scaling studies.
- The paper is extremely detailed, providing extensive explanations of both the dry and wet lab experiments.

Weaknesses:
- The use of IRPO/preference learning for supervised fitness prediction seems poorly motivated -- since IRPO (and most offline preference learning algorithms) trains on ranked pairs, using this algorithm necessarily involves some loss of information (i.e., the magnitudes of the experimentally-determined fitness values). Additionally, such preference learning algorithms have been shown to struggle with learning likelihoods that correctly align with the rankings of the data (see [Chen et al.](https://arxiv.org/abs/2405.19534)). Although it is trendy now to use IRPO and other rankings-based preference learning methods for fine-tuning pLMs, using it for a regression task is not very principled. (Additionally, it is an unfortunate fact that many popular offline preference learning algorithms still show a non-monotonic relationship between ranking performance and generality quality, so it may be better to not try to optimize for both at once with these algorithms. See [Tang et al.](https://arxiv.org/abs/2405.08448) and Sec. 5 of [Chen et al.](https://arxiv.org/abs/2405.19534) for evidence of this.)
- Very minor nits:
  - I find figure 1(a) to be somewhat incomprehensible. Even as someone who already understands what infilling generation is, I still did not understand this diagram.
  - Typo in Fig. 3 caption: "generated generated" -> "generated"

---

> ### Author Rebuttal · Authors · 2025-07-31
>
> Thank you for your thoughtful feedback and the highly relevant references. We did some early experiments with offline preference learning algorithms like LiPO [1] that make full use of the continuous fitness values, but we found that they generally underperformed DPO/IRPO. Prior work has also shown that contrastive ranking losses can be more effective than regression losses for protein variant effect prediction, especially in few-shot settings like those we consider in Figure 4b [2-3]. However, we agree that moving beyond ranked pairs is an important direction for future research in aligning protein language models with continuous-valued experimental data.
>
> Your points about the gap between predictive & generative capabilities and between online & offline algorithms are interesting ones. We do find that aligning on the Megascale stability dataset allows us to generate proteins that are more stable both computationally (Figure 4f) and experimentally (Figure 4g); additionally, larger models generate more stable proteins than smaller models (Figure 4f), matching the trends of the stability prediction results (Figure 4c-e). A key difference between our setting and the NLP one is that we select <100 low-perplexity sequences out of 50,000+ generated candidates, while in NLP, LLMs rarely generate >100 candidates. Our setting may therefore increase the importance of the model’s ranking capabilities and mitigate some of its generative shortcomings.
>
> We chose not to use an online algorithm due to the additional complexities it introduces: we would need to train a different reward model for each dataset, ensure that reward model generalized from the local mutational landscapes it was trained on to model generations, and fine-tune and/or prompt ProGen3 for each dataset to ensure that we sample generations from the right protein cluster. Given that we trained 4 models on 9 datasets each, it was impractical to fully tune all these hyperparameters for each setting, but comparing online and offline methods (especially in settings where the generations from each can be tested in the lab) is an exciting future direction.
>
> We are more than happy to address your minor points, especially if you have any specific feedback on what would make Figure 1a more interpretable.
>
> [1] Liu et al. “LiPO: Listwise Preference Optimization through Learning-to-Rank.” NAACL, 2025.
>
> [2] Brookes et al. “Contrastive losses as generalized models of global epistasis.” NeurIPS, 2024.
>
> [3] Zhao et al. “Contrastive Fitness Learning: Reprogramming Protein Language Models for Low-N Learning of Protein Fitness Landscape.” RECOMB, 2024.

---

> > ### Author Response · Authors · 2025-08-06
> >
> > Thank you for your review. Please let us know if our comments addressed your concerns, or if you have any outstanding questions.

---

### Official Review · Reviewer_GjZd · 2025-07-03

**Clarity:** 4
**Significance:** 4
**Originality:** 3
**Rating:** 5
**Confidence:** 4

**Summary:**

This paper presents ProGen3, a protein LLM scaled up to 46 billion parameters.  The paper goes into significant depth on the development of a new training dataset and its impacts on performance as the number of parameters/data is scaled.  The paper then proceeds with significant evaluation of the ProGen3 family of models, finding that as parameters scale the model is able to generate more diverse protein sequences.  Samples from ProGen3 are evaluated in the wet lab where it is demonstrated that the generated proteins express.  Finally, the authors perform preference alignment on the ProGen3 model and demonstrate that scaling impacts model performance after fine-tuning.

**Questions:**

1. What are the authors' hunches for why the largest models don't lead to better zero-shot fitness prediction?

2. Similarly, what are the authors' intuitions for why scaling model size doesn't seem to impact expression rates all that much besides on the infilling task?  Also, why do the authors think scaling improves infilling performance in particular?

3. Will model weights for the 46B parameter model as well as the training dataset be open-sourced?

**Ethical Concerns:**

["NO or VERY MINOR ethics concerns only"]

**Final Justification:**

The paper is well motivated, delves into the engineering of training a scaled PLM, and shows impressive wet lab results.  As such, I recommend acceptance for this paper.

**Limitations:**

Yes

**Quality:**

4

**Strengths And Weaknesses:**

This paper proposes ProGen3 and details its training procedures (including a new training dataset) and goes into great depth on analysis of the impact of scaling on downstream performance.  The writing is clear and concise and the findings are quite comprehensive.  I would like to in particular commend the detail in this paper on the specifics of scaling up ProGen3 and the various design factors which impacted downstream performance.  Moreover, the evaluation of the model was very comprehensive, covering many different axes of model performance, including wet lab validation.

I do have one concern regarding the reproducibility of the experiments detailed in the paper.  While ProGen3 model weights have been released on HuggingFace for model sizes up to 3B, the 46B parameter model has not been open sourced.  Further, as training set mixes and setups are one of the key drivers of performance in LLM training, it would be very beneficial if the PPA-1 training set described in the manuscript were open sourced, especially as small discrepancies in training sets can cause large impacts on downstream model performance.  I would strongly encourage the authors to consider open sourcing both of these to better the reproducibility of their work, as training a 46B model on 256 H100s for 17 days is impractical for nearly any academic lab.

---

> ### Author Rebuttal · Authors · 2025-07-31
>
> Thank you for your thoughtful feedback. Addressing your points one at a time,
> 1. As you mentioned, we have open sourced all model sizes up to 3B parameters, but we do not intend to open source PPA-1 or ProGen3-46B.
> 2. As models grow larger, they improve at estimating the distribution of naturally occurring proteins. However, since the forces driving evolution don’t always correlate with the experimental read-outs of laboratory assays, the average protein found in nature is rarely the most optimal. Thus, effectively modeling the evolutionary distribution of proteins can be at odds with zero-shot fitness prediction, since the most optimal variants may be quite unlikely in nature. Others have also theorized about and documented this trend [1-2], and we allude to their work in Lines 219-226. Rather, we find that larger models learn better representations that are more useful for downstream tasks.
> 3. Based on our findings, we expect that it is actually quite easy for ProGen3 models to generate expressing (but potentially non-functional) proteins, *provided that those proteins come from families that are in-distribution for the model*. Consequently, unconditional generations from all models have similar expression rates when compared head-to-head, but larger models trained on more data can generate expressing proteins for families that are out-of-distribution for the smaller models. For the infilling task, we pre-selected protein scaffolds for the model to infill, and a number of these scaffolds happened to be in-distribution for ProGen3-46B but out-of-distribution for ProGen3-3B (but not vice versa).
>
> [1] Weinstein et al. "Non-identifiability and the blessings of misspecification in models of molecular fitness.” NeurIPS, 2022.
>
> [2] Gordon et al. “Protein language model fitness is a matter of preference.” ICLR, 2025.

---

### Decision · Program_Chairs · 2025-09-17

**Decision:**

Accept (spotlight)

**Comment:**

This work presents ProGen3, a family of protein language models up to 46B parameters. Reviewers appreciated the extensive scaling analysis, creation of a new dataset, and compelling wet-lab validation. All reviewers recommended acceptance, agreeing that the paper's contributions are significant and its strengths outweigh its weaknesses.

The paper is a clear accept. The scale of the models combined with a comprehensive evaluation that includes wet-lab validation makes this a useful study in protein generation. The insights on scaling and data curation are a contribution to the field. While the reproducibility concerns are valid, they do not detract from the paper's overall impact and the value of the released models.